# Binary Classification with Confidence Difference

**Wei Wang**[1,2], **Lei Feng**[3,2], **Yuchen Jiang**[4], **Gang Niu**[2], **Min-Ling Zhang**[5], **Masashi Sugiyama**[2,1]

[1] The University of Tokyo, Chiba, Japan
[2] RIKEN Center for Advanced Intelligence Project, Tokyo, Japan
[3] Nanyang Technological University, Singapore
[4] Alibaba Group, Beijing, China
[5] Southeast University, Nanjing, China

wangw@g.ecc.u-tokyo.ac.jp    lfengqaq@gmail.com
jiangyuchen.jyc@alibaba-inc.com    gang.niu.ml@gmail.com
zhangml@seu.edu.cn    sugi@k.u-tokyo.ac.jp

## Abstract

Recently, learning with *soft labels* has been shown to achieve better performance than learning with *hard labels* in terms of model generalization, calibration, and robustness. However, collecting pointwise labeling confidence for all training examples can be challenging and time-consuming in real-world scenarios. This paper delves into a novel weakly supervised binary classification problem called *confidence-difference (ConfDiff) classification*. Instead of pointwise labeling confidence, we are given only unlabeled data pairs with confidence difference that specifies the difference in the probabilities of being positive. We propose a risk-consistent approach to tackle this problem and show that the estimation error bound achieves the optimal convergence rate. We also introduce a risk correction approach to mitigate overfitting problems, whose consistency and convergence rate are also proven. Extensive experiments on benchmark data sets and a real-world recommender system data set validate the effectiveness of our proposed approaches in exploiting the supervision information of the confidence difference.

## 1 Introduction

Recent years have witnessed the prevalence of deep learning and its successful applications. However, the success is built on the basis of the collection of large amounts of data with unique and accurate labels. However, in many real-world scenarios, it is often difficult to satisfy such requirements. To circumvent the difficulty, various weakly supervised learning problems have been investigated accordingly, including but not limited to semi-supervised learning [1, 2, 3, 4], label-noise learning [5, 6, 7, 8, 9], positive-unlabeled learning [10, 11, 12], partial-label learning [13, 14, 15, 16, 17], unlabeled-unlabeled learning [18, 19], and similarity-based classification [20, 21, 22].

Learning with soft labels has been shown to achieve better performance than learning with hard labels in the context of supervised learning [23, 24], where each example is equipped with *pointwise labeling confidence* indicating the degree to which the labels describe the example. The advantages have been validated in many aspects, including model generalization [25, 26], calibration [27, 28], and robustness [29, 30]. For example, with the help of soft labels, knowledge distillation [25, 31] transfers knowledge from a large teacher network to a small student network. The student network can be trained more efficiently and reliably with the soft labels generated by the teacher network [32, 33, 34].

However, collecting a large number of training examples with pointwise labeling confidence may be demanding under many circumstances since it is challenging to describe the labeling confidence for each training example exactly [35, 36, 37]. Different annotators may give different values of pointwise

labeling confidence to the same example due to personal biases, and it has been demonstrated that skewed confidence values can harm classification performance [36]. Besides, giving pointwise labeling information to large-scale data sets is also expensive, laborious, and even unrealistic in many real-world scenarios [38, 39, 9]. On the contrary, leveraging supervision information of pairwise comparisons may ameliorate the biases of skewed pointwise labeling confidence and save labeling costs. Following this idea, we investigate a more practical problem setting for binary classification in this paper, where we are given *unlabeled data pairs with confidence difference* indicating the difference in the probabilities of being positive. Collecting confidence difference for training examples in pairs is much cheaper and more accessible than collecting pointwise labeling confidence for all the training examples.

Take click-through rate prediction in recommender systems [40, 41] for example. The combinations of users and their favorite/disliked items can be regarded as positive/negative data. Collecting training data takes work to distinguish between positive and negative data. Furthermore, the pointwise labeling confidence of training data may be difficult to be determined due to the extraordinarily sparse and class-imbalance problems [42]. Therefore, the collected confidence values may be biased. However, collecting the difference in the preference between a pair of candidate items for a given user is more accessible and may alleviate the biases. Section 4 will give a real-world case study of this problem. Take the disease risk estimation problem for another example. Given a person's attributes, the goal is to predict the risk of having some disease. When asking doctors to annotate the probabilities of having the disease for patients, it takes work to determine the exact values of the probabilities. Furthermore, the probability values given by different doctors may differ due to their diverse backgrounds. On the other hand, it is much easier and less biased to estimate the relative difference in the probabilities of having the disease between two patients. Therefore, the problem of learning with confidence difference is of practical research value, but has yet to be investigated in the literature.

As a related work, Feng et al. [43] elaborated that a binary classifier can be learned from pairwise comparisons, termed Pcomp classification. For a pair of samples, they used a pairwise label of one is being more likely to be positive than the other. Since knowing the confidence difference implies knowing the pairwise label, our method requires stronger supervision than Pcomp classification. Nevertheless, we argue that, in many real-world scenarios, we may not only know one example is more likely to be positive than the other, but also know how much the difference in confidence is, as explained in the above examples. Therefore, the setting of the current paper is not so restrictive compared with that of Pcomp classification. Furthermore, our setting is more flexible than that of Pcomp classification from the viewpoint of data generation process. Pcomp classification limits the labels of pairs of training data to be in $\{(+1, +1), (+1, -1), (-1, -1)\}$. To cope with collected data with labels $(-1, +1)$, they either discard them or reverse them as $(+1, -1)$. On the contrary, our setting is more general and we take the examples from $(-1, +1)$ also into consideration explicitly in the data distribution assumption. Figure 1 shows the results of a pilot experiment. Here,

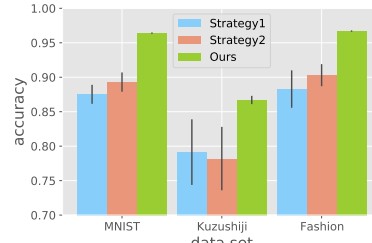

Figure 1: Experimental results for different strategies of handling examples from $(-1, +1)$.

Strategy1 denotes Pcomp-Teacher [43] discarding examples from $(-1, +1)$ while Strategy2 denotes Pcomp-Teacher reversing examples from $(-1, +1)$. We can observe that both strategies perform actually similarly. Since Strategy1 loses many data, it is intuitively expected that Strategy2 works much better, but the improvement is actually marginal. This may be because the training data distribution after reversing differs from the expected one. On the contrary, in this work we take examples from $(-1, +1)$ into consideration directly, which is a more appropriate way to handle these data.

Learning with pairwise comparisons has been investigated pervasively in the community [44, 45, 46, 47, 48, 49, 50], with applications in information retrieval [51], computer vision [52], regression [53, 54], crowdsourcing [55, 56], and graph learning [57]. It is noteworthy that there exist distinct differences between our work and previous works on learning with pairwise comparisons. Previous works have mainly tried to learn a ranking function that can rank candidate examples according to relevance or preference. In this paper, we try to learn a *pointwise binary classifier* by conducting *empirical risk minimization* under the binary classification setting.

Our contributions are summarized as follows:

- We investigate confidence-difference (ConfDiff) classification, a novel and practical weakly supervised learning problem, which can be solved via empirical risk minimization by constructing an *unbiased risk estimator*. The proposed approach can be equipped with any model, loss function, and optimizer flexibly.
- An estimation error bound is derived, showing that the proposed approach achieves the optimal parametric convergence rate. The robustness is further demonstrated by probing into the influence of an inaccurate class prior probability and noisy confidence difference.
- To mitigate overfitting issues, a risk correction approach [19] with consistency guarantee is further introduced. Extensive experimental results on benchmark data sets and a real-world recommender system data set validate the effectiveness of the proposed approaches.

## 2 Preliminaries

In this section, we discuss the background of binary classification, binary classification with soft labels, and Pcomp classification. Then, we elucidate the data generation process of ConfDiff classification.

### 2.1 Binary Classification

For binary classification, let $\mathcal{X} = \mathbb{R}^d$ denote the $d$-dimensional feature space and $\mathcal{Y} = \{+1, -1\}$ denote the label space. Let $p(\boldsymbol{x}, y)$ denote the unknown joint probability density over random variables $(\boldsymbol{x}, y) \in \mathcal{X} \times \mathcal{Y}$. The task of binary classification is to learn a binary classifier $g : \mathcal{X} \to \mathbb{R}$ which minimizes the following classification risk:

$$R(g) = \mathbb{E}_{p(\boldsymbol{x}, y)}[\ell(g(\boldsymbol{x}), y)], \tag{1}$$

where $\ell(\cdot, \cdot)$ is a non-negative binary-class loss function, such as the 0-1 loss and logistic loss. Let $\pi_+ = p(y = +1)$ and $\pi_- = p(y = -1)$ denote the class prior probabilities for the positive and negative classes respectively. Furthermore, let $p_+(\boldsymbol{x}) = p(\boldsymbol{x}|y = +1)$ and $p_-(\boldsymbol{x}) = p(\boldsymbol{x}|y = -1)$ denote the class-conditional probability densities of positive and negative data respectively. Then the classification risk in Eq. (1) can be equivalently expressed as

$$R(g) = \pi_+ \mathbb{E}_{p_+(\boldsymbol{x})}[\ell(g(\boldsymbol{x}), +1)] + \pi_- \mathbb{E}_{p_-(\boldsymbol{x})}[\ell(g(\boldsymbol{x}), -1)]. \tag{2}$$

### 2.2 Binary Classification with Soft Labels

When the soft labels of training examples are accessible to the learning algorithm, taking advantage of them can often improve the generalization performance [23]. First, the classification risk in Eq. (1) can be equivalently expressed as

$$R(g) = \mathbb{E}_{p(\boldsymbol{x})}[p(y = +1|\boldsymbol{x})\ell(g(\boldsymbol{x}), +1) + p(y = -1|\boldsymbol{x})\ell(g(\boldsymbol{x}), -1)]. \tag{3}$$

Then, given training data equipped with confidence $\{(\boldsymbol{x}_i, r_i)\}_{i=1}^n$ where $r_i = p(y_i = +1|\boldsymbol{x}_i)$ is the *pointwise positive confidence* associated with $\boldsymbol{x}_i$, we minimize the following unbiased risk estimator to perform empirical risk minimization:

$$\widehat{R}_{\text{soft}}(g) = \frac{1}{n} \sum\nolimits_{i=1}^n (r_i \ell(g(\boldsymbol{x}_i), +1) + (1 - r_i)\ell(g(\boldsymbol{x}_i), -1)). \tag{4}$$

However, accurate pointwise positive confidence may be hard to be obtained in reality [36].

### 2.3 Pairwise-Comparison (Pcomp) Classification

In principle, collecting supervision information of pairwise comparisons is much easier and cheaper than pointwise supervision information [43]. In Pcomp classification [43], we are given pairs of unlabeled data where we know which one is more likely to be positive than the other. It is assumed that Pcomp data are sampled from labeled data pairs whose labels belong to $\{(+1, -1), (+1, +1), (-1, -1)\}$. Based on this assumption, the probability density of Pcomp data $(\boldsymbol{x}, \boldsymbol{x}')$ is given as $q(\boldsymbol{x}, \boldsymbol{x}')/(\pi_+^2 + \pi_-^2 + \pi_+\pi_-)$ where $q(\boldsymbol{x}, \boldsymbol{x}') = \pi_+^2 p_+(\boldsymbol{x})p_+(\boldsymbol{x}') + \pi_-^2 p_-(\boldsymbol{x})p_-(\boldsymbol{x}') + \pi_+\pi_- p_+(\boldsymbol{x})p_-(\boldsymbol{x}')$. Then, an unbiased risk estimator for Pcomp classification is derived as follows:

$$\widehat{R}_{\text{Pcomp}}(g) = \frac{1}{n} \sum\nolimits_{i=1}^n (\ell(g(\boldsymbol{x}_i), +1) + \ell(g(\boldsymbol{x}_i'), -1) - \pi_+\ell(g(\boldsymbol{x}_i), -1) - \pi_-\ell(g(\boldsymbol{x}_i'), +1)). \tag{5}$$

In real-world scenarios, we may not only know one example is more likely to be positive than the other, but also know how much the difference in confidence is. Next, a novel weakly supervised learning setting named ConfDiff classification is introduced which can utilize such confidence difference.

## 2.4 Confidence-Difference (ConfDiff) Classification

In this subsection, the formal definition of confidence difference is given firstly. Then, we elaborate the data generation process of ConfDiff data.

**Definition 1** (Confidence Difference). *The confidence difference $c(\boldsymbol{x}, \boldsymbol{x}')$ between an unlabeled data pair $(\boldsymbol{x}, \boldsymbol{x}')$ is defined as*

$$c(\boldsymbol{x}, \boldsymbol{x}') = p(y' = +1 | \boldsymbol{x}') - p(y = +1 | \boldsymbol{x}). \tag{6}$$

As shown in the definition above, the confidence difference denotes the difference in the class posterior probabilities between the unlabeled data pair, which can measure how confident the pairwise comparison is. In ConfDiff classification, we are only given $n$ unlabeled data pairs with confidence difference $\mathcal{D} = \{((\boldsymbol{x}_i, \boldsymbol{x}'_i), c_i)\}_{i=1}^n$. Here, $c_i = c(\boldsymbol{x}_i, \boldsymbol{x}'_i)$ is the confidence difference for the unlabeled data pair $(\boldsymbol{x}_i, \boldsymbol{x}'_i)$. Furthermore, the unlabeled data pair $(\boldsymbol{x}_i, \boldsymbol{x}'_i)$ is assumed to be drawn from a probability density $p(\boldsymbol{x}, \boldsymbol{x}') = p(\boldsymbol{x})p(\boldsymbol{x}')$. This indicates that $\boldsymbol{x}_i$ and $\boldsymbol{x}'_i$ are two i.i.d. instances sampled from $p(\boldsymbol{x})$. It is worth noting that the confidence difference $c_i$ will be positive if the second instance $\boldsymbol{x}'_i$ has a higher probability to be positive than the first instance $\boldsymbol{x}_i$, and will be negative otherwise. During the data collection process, the labeler can first sample two unlabeled data independently from the marginal distribution $p(\boldsymbol{x})$, then provide the confidence difference for them.

# 3 The Proposed Approach

In this section, we introduce our proposed approaches with theoretical guarantees. Besides, we show the influence of an inaccurate class prior probability and noisy confidence difference theoretically. Furthermore, we introduce a risk correction approach to improve the generalization performance.

## 3.1 Unbiased Risk Estimator

In this subsection, we show that the classification risk in Eq. (1) can be expressed with ConfDiff data in an equivalent way.

**Theorem 1.** *The classification risk $R(g)$ in Eq. (1) can be equivalently expressed as*

$$R_{\mathrm{CD}}(g) = \mathbb{E}_{p(\boldsymbol{x}, \boldsymbol{x}')}[\frac{1}{2}(\mathcal{L}(\boldsymbol{x}, \boldsymbol{x}') + \mathcal{L}(\boldsymbol{x}', \boldsymbol{x}))], \tag{7}$$

*where*

$$\mathcal{L}(\boldsymbol{x}, \boldsymbol{x}') = (\pi_+ - c(\boldsymbol{x}, \boldsymbol{x}'))\ell(g(\boldsymbol{x}), +1) + (\pi_- - c(\boldsymbol{x}, \boldsymbol{x}'))\ell(g(\boldsymbol{x}'), -1).$$

Accordingly, we can derive an unbiased risk estimator for ConfDiff classification:

$$\widehat{R}_{\mathrm{CD}}(g) = \frac{1}{2n} \sum_{i=1}^n (\mathcal{L}(\boldsymbol{x}_i, \boldsymbol{x}'_i) + \mathcal{L}(\boldsymbol{x}'_i, \boldsymbol{x}_i)). \tag{8}$$

**Minimum-variance risk estimator.** Actually, Eq. (8) is one of the candidates of the unbiased risk estimator. We introduce the following lemma:

**Lemma 1.** *The following expression is also an unbiased risk estimator:*

$$\frac{1}{n} \sum_{i=1}^n (\alpha \mathcal{L}(\boldsymbol{x}_i, \boldsymbol{x}'_i) + (1 - \alpha)\mathcal{L}(\boldsymbol{x}'_i, \boldsymbol{x}_i)), \tag{9}$$

*where $\alpha \in [0, 1]$ is an arbitrary weight.*

Then, we introduce the following theorem:

**Theorem 2.** *The unbiased risk estimator in Eq. (8) has the minimum variance among all the candidate unbiased risk estimators in the form of Eq. (9) w.r.t. $\alpha \in [0, 1]$.*

Theorem 2 indicates the variance minimality of the proposed unbiased risk estimator in Eq. (8), and we adopt this risk estimator in the following sections.

## 3.2 Estimation Error Bound

In this subsection, we elaborate the convergence property of the proposed risk estimator $\widehat{R}_{\mathrm{CD}}(g)$ by giving an estimation error bound. Let $\mathcal{G} = \{g : \mathcal{X} \mapsto \mathbb{R}\}$ denote the model class. It is assumed that there exists some constant $C_g$ such that $\sup_{g \in \mathcal{G}} \|g\|_\infty \leq C_g$ and some constant $C_\ell$ such that $\sup_{|z| \leq C_g} \ell(z, y) \leq C_\ell$. We also assume that the binary loss function $\ell(z, y)$ is Lipschitz continuous for $z$ and $y$ with a Lipschitz constant $L_\ell$. Let $g^* = \arg\min_{g \in \mathcal{G}} R(g)$ denote the minimizer of the classification risk in Eq. (1) and $\widehat{g}_{\mathrm{CD}} = \arg\min_{g \in \mathcal{G}} \widehat{R}_{\mathrm{CD}}(g)$ denote the minimizer of the unbiased risk estimator in Eq. (8). The following theorem can be derived:

**Theorem 3.** *For any $\delta > 0$, the following inequality holds with probability at least $1 - \delta$:*

$$R(\widehat{g}_{\mathrm{CD}}) - R(g^*) \leq 8L_\ell \mathfrak{R}_n(\mathcal{G}) + 4C_\ell \sqrt{\frac{\ln 2/\delta}{2n}}, \tag{10}$$

*where $\mathfrak{R}_n(\mathcal{G})$ denotes the Rademacher complexity of $\mathcal{G}$ for unlabeled data with size $n$.*

From Theorem 3, we can observe that as $n \to \infty$, $R(\widehat{g}_{\mathrm{CD}}) \to R(g^*)$ because $\mathcal{R}_n(\mathcal{G}) \to 0$ for all parametric models with a bounded norm, such as deep neural networks trained with weight decay [58]. Furthermore, the estimation error bound converges in $\mathcal{O}_p(1/\sqrt{n})$, where $\mathcal{O}_p$ denotes the order in probability, which is the optimal parametric rate for empirical risk minimization without making additional assumptions [59].

## 3.3 Robustness of Risk Estimator

In the previous subsections, it was assumed that the class prior probability is known in advance. In addition, it was assumed that the ground-truth confidence difference of each unlabeled data pair is accessible. However, these assumptions can rarely be satisfied in real-world scenarios, since the collection of confidence difference is inevitably injected with noise. In this subsection, we theoretically analyze the influence of an inaccurate class prior probability and noisy confidence difference on the learning procedure. Later in Section 4.5, we will experimentally verify our theoretical findings.

Let $\bar{\mathcal{D}} = \{((\boldsymbol{x}_i, \boldsymbol{x}_i'), \bar{c}_i)\}_{i=1}^n$ denote $n$ unlabeled data pairs with noisy confidence difference, where $\bar{c}_i$ is generated by corrupting the ground-truth confidence difference $c_i$ with noise. Besides, let $\bar{\pi}_+$ denote the inaccurate class prior probability accessible to the learning algorithm. Furthermore, let $\bar{R}_{\mathrm{CD}}(g)$ denote the empirical risk calculated based on the inaccurate class prior probability and noisy confidence difference. Let $\bar{g}_{\mathrm{CD}} = \arg\min_{g \in \mathcal{G}} \bar{R}_{\mathrm{CD}}(g)$ denote the minimizer of $\bar{R}_{\mathrm{CD}}(g)$. Then the following theorem gives an estimation error bound:

**Theorem 4.** *Based on the assumptions above, for any $\delta > 0$, the following inequality holds with probability at least $1 - \delta$:*

$$R(\bar{g}_{\mathrm{CD}}) - R(g^*) \leq 16L_\ell \mathfrak{R}_n(\mathcal{G}) + 8C_\ell \sqrt{\frac{\ln 2/\delta}{2n}} + \frac{4C_\ell \sum_{i=1}^n |\bar{c}_i - c_i|}{n} + 4C_\ell |\bar{\pi}_+ - \pi_+|. \tag{11}$$

Theorem 4 indicates that the estimation error is bounded by twice the original bound in Theorem 3 with the mean absolute error of the noisy confidence difference and the inaccurate class prior probability. Furthermore, if $\sum_{i=1}^n |\bar{c}_i - c_i|$ has a sublinear growth rate with high probability and the class prior probability is estimated consistently, the risk estimator can be even consistent. It elaborates the robustness of the proposed approach.

## 3.4 Risk Correction Approach

It is worth noting that the empirical risk in Eq. (8) may be negative due to negative terms, which is unreasonable because of the non-negative property of loss functions. This phenomenon will result in severe overfitting problems when complex models are adopted [19, 22, 43]. To circumvent this difficulty, we wrap the individual loss terms in Eq. (8) with *risk correction functions* proposed in Lu et al. [19], such as the rectified linear unit (ReLU) function $f(z) = \max(0, z)$ and the absolute value function $f(z) = |z|$. In this way, the corrected risk estimator for ConfDiff classification can be

expressed as follows:

$$\widetilde{R}_{\mathrm{CD}}(g) = \frac{1}{2n}(f(\sum_{i=1}^{n}(\pi_+ - c_i)\ell(g(\boldsymbol{x}_i), +1)) + f(\sum_{i=1}^{n}(\pi_- - c_i)\ell(g(\boldsymbol{x}_i'), -1))$$

$$+ f(\sum_{i=1}^{n}(\pi_+ + c_i)\ell(g(\boldsymbol{x}_i'), +1)) + f(\sum_{i=1}^{n}(\pi_- + c_i)\ell(g(\boldsymbol{x}_i), -1))). \quad (12)$$

**Theoretical analysis.** We assume that the risk correction function $f(z)$ is Lipschitz continuous with Lipschitz constant $L_f$. For ease of notation, let $\widehat{A}(g) = \sum_{i=1}^{n}(\pi_+ - c_i)\ell(g(\boldsymbol{x}_i), +1)/2n$, $\widehat{B}(g) = \sum_{i=1}^{n}(\pi_- - c_i)\ell(g(\boldsymbol{x}_i'), -1)/2n$, $\widehat{C}(g) = \sum_{i=1}^{n}(\pi_+ + c_i)\ell(g(\boldsymbol{x}_i'), +1)/2n$, and $\widehat{D}(g) = \sum_{i=1}^{n}(\pi_- + c_i)\ell(g(\boldsymbol{x}_i), -1)/2n$. From Lemma 3 in Appendix A, the values of $\mathbb{E}[\widehat{A}(g)], \mathbb{E}[\widehat{B}(g)], \mathbb{E}[\widehat{C}(g)]$, and $\mathbb{E}[\widehat{D}(g)]$ are non-negative. Therefore, we assume that there exist non-negative constants $a, b, c$, and $d$ such that $\mathbb{E}[\widehat{A}(g)] \geq a, \mathbb{E}[\widehat{B}(g)] \geq b, \mathbb{E}[\widehat{C}(g)] \geq c$, and $\mathbb{E}[\widehat{D}(g)] \geq d$. Besides, let $\widetilde{g}_{\mathrm{CD}} = \arg\min_{g \in \mathcal{G}} \widetilde{R}_{\mathrm{CD}}(g)$ denote the minimizer of $\widetilde{R}_{\mathrm{CD}}(g)$. Then, Theorem 5 is provided to elaborate the bias and consistency of $\widetilde{R}_{\mathrm{CD}}(g)$.

**Theorem 5.** *Based on the assumptions above, the bias of the risk estimator $\widetilde{R}_{\mathrm{CD}}(g)$ decays exponentially as $n \to \infty$:*

$$0 \leq \mathbb{E}[\widetilde{R}_{\mathrm{CD}}(g)] - R(g) \leq 2(L_f + 1)C_\ell\Delta, \quad (13)$$

*where $\Delta = \exp\left(-2a^2 n/C_\ell^2\right) + \exp\left(-2b^2 n/C_\ell^2\right) + \exp\left(-2c^2 n/C_\ell^2\right) + \exp\left(-2d^2 n/C_\ell^2\right)$. Furthermore, with probability at least $1 - \delta$, we have*

$$|\widetilde{R}_{\mathrm{CD}}(g) - R(g)| \leq 2C_\ell L_f \sqrt{\frac{\ln 2/\delta}{2n}} + 2(L_f + 1)C_\ell\Delta. \quad (14)$$

Theorem 5 demonstrates that $\widetilde{R}_{\mathrm{CD}}(g) \to R(g)$ in $\mathcal{O}_p(1/\sqrt{n})$, which means that $\widetilde{R}_{\mathrm{CD}}(g)$ is biased yet consistent. The estimation error bound of $\widetilde{g}_{\mathrm{CD}}$ is analyzed in Theorem 6.

**Theorem 6.** *Based on the assumptions above, for any $\delta > 0$, the following inequality holds with probability at least $1 - \delta$:*

$$R(\widetilde{g}_{\mathrm{CD}}) - R(g^*) \leq 8L_\ell\mathfrak{R}_n(\mathcal{G}) + 4C_\ell(L_f + 1)\sqrt{\frac{\ln 2/\delta}{2n}} + 4(L_f + 1)C_\ell\Delta. \quad (15)$$

Theorem 6 elucidates that as $n \to \infty$, $R(\widetilde{g}_{\mathrm{CD}}) \to R(g^*)$, since $\mathcal{R}_n(\mathcal{G}) \to 0$ for all parametric models with a bounded norm [60] and $\Delta \to 0$. Furthermore, the estimation error bound converges in $\mathcal{O}_p(1/\sqrt{n})$, which is the optimal parametric rate for empirical risk minimization without additional assumptions [59].

## 4  Experiments

In this section, we verify the effectiveness of our proposed approaches experimentally.

### 4.1  Experimental Setup

We conducted experiments on benchmark data sets, including MNIST [61], Kuzushiji-MNIST [62], Fashion-MNIST [63], and CIFAR-10 [64]. In addition, four UCI data sets [65] were used, including Optdigits, USPS, Pendigits, and Letter. Since the data sets were originally designed for multi-class classification, we manually partitioned them into binary classes. For CIFAR-10, we used ResNet-34 [66] as the model architecture. For other data sets, we used a multilayer perceptron (MLP) with three hidden layers of width 300 equipped with the ReLU [67] activation function and batch normalization [68]. The logistic loss is utilized to instantiate the loss function $\ell$.

It is worth noting that confidence difference is given by labelers in real-world applications, while it was generated synthetically in this paper to facilitate comprehensive experimental analysis. We firstly trained a probabilistic classifier via logistic regression with ordinarily labeled data and the same neural network architecture. Then, we sampled unlabeled data in pairs at random, and generated the class posterior probabilities by inputting them into the probabilistic classifier. After that, we

Table 1: Classification accuracy (mean±std) of each method on benchmark data sets with different class priors, where the best performance (excluding Oracle) is shown in bold.

| Class Prior | Method | MNIST | Kuzushiji | Fashion | CIFAR-10 |
|---|---|---|---|---|---|
| $\pi_+ = 0.2$ | Pcomp-Unbiased | 0.761±0.017 | 0.637±0.052 | 0.737±0.050 | 0.776±0.023 |
| | Pcomp-ReLU | 0.800±0.000 | 0.800±0.000 | 0.800±0.000 | 0.800±0.000 |
| | Pcomp-ABS | 0.800±0.000 | 0.800±0.000 | 0.800±0.000 | 0.800±0.000 |
| | Pcomp-Teacher | 0.965±0.010 | 0.871±0.046 | 0.853±0.017 | 0.836±0.019 |
| | Oracle-Hard | 0.990±0.000 | 0.939±0.001 | 0.979±0.001 | 0.894±0.003 |
| | Oracle-Soft | 0.989±0.001 | 0.939±0.004 | 0.979±0.001 | 0.893±0.003 |
| | ConfDiff-Unbiased | 0.789±0.041 | 0.672±0.053 | 0.855±0.024 | 0.789±0.025 |
| | ConfDiff-ReLU | 0.968±0.003 | 0.860±0.017 | 0.964±0.004 | 0.844±0.020 |
| | ConfDiff-ABS | **0.975±0.003** | **0.898±0.003** | **0.965±0.002** | **0.862±0.015** |
| $\pi_+ = 0.5$ | Pcomp-Unbiased | 0.712±0.020 | 0.578±0.036 | 0.723±0.042 | 0.703±0.042 |
| | Pcomp-ReLU | 0.502±0.003 | 0.502±0.004 | 0.500±0.000 | 0.602±0.032 |
| | Pcomp-ABS | 0.842±0.012 | 0.727±0.006 | 0.851±0.012 | 0.583±0.018 |
| | Pcomp-Teacher | 0.893±0.014 | 0.782±0.046 | 0.903±0.016 | 0.779±0.016 |
| | Oracle-Hard | 0.986±0.000 | 0.929±0.002 | 0.976±0.001 | 0.871±0.003 |
| | Oracle-Soft | 0.985±0.001 | 0.928±0.002 | 0.978±0.001 | 0.877±0.002 |
| | ConfDiff-Unbiased | 0.911±0.046 | 0.712±0.046 | 0.896±0.036 | 0.720±0.024 |
| | ConfDiff-ReLU | 0.944±0.011 | 0.805±0.015 | 0.960±0.003 | 0.830±0.007 |
| | ConfDiff-ABS | **0.964±0.001** | **0.867±0.006** | **0.967±0.001** | **0.843±0.004** |
| $\pi_+ = 0.8$ | Pcomp-Unbiased | 0.799±0.005 | 0.671±0.029 | 0.813±0.029 | 0.737±0.022 |
| | Pcomp-ReLU | 0.910±0.031 | 0.775±0.022 | 0.897±0.023 | 0.851±0.010 |
| | Pcomp-ABS | 0.854±0.027 | 0.838±0.026 | 0.921±0.017 | 0.849±0.007 |
| | Pcomp-Teacher | 0.943±0.026 | 0.814±0.027 | 0.936±0.014 | 0.821±0.003 |
| | Oracle-Hard | 0.991±0.001 | 0.942±0.003 | 0.979±0.000 | 0.897±0.002 |
| | Oracle-Soft | 0.990±0.002 | 0.945±0.003 | 0.980±0.001 | 0.904±0.009 |
| | ConfDiff-Unbiased | 0.792±0.017 | 0.758±0.033 | 0.810±0.035 | 0.794±0.012 |
| | ConfDiff-ReLU | 0.970±0.004 | 0.886±0.009 | 0.970±0.002 | 0.851±0.012 |
| | ConfDiff-ABS | **0.983±0.002** | **0.915±0.001** | **0.975±0.002** | **0.874±0.011** |

generated confidence difference for each pair of sampled data according to Definition 1. To verify the effectiveness of our approaches under different class prior settings, we set $\pi_+ \in \{0.2, 0.5, 0.8\}$ for all the data sets. Besides, we assumed that the class prior $\pi_+$ was known for all the compared methods. We repeated the sampling-and-training procedure for five times, and the mean accuracy as well as the standard deviation were recorded.

We adopted the following variants of our proposed approaches: 1) ConfDiff-Unbiased, which denotes the method working by minimizing the unbiased risk estimator; 2) ConfDiff-ReLU, which denotes the method working by minimizing the corrected risk estimator with the ReLU function as the risk correction function; 3) ConfDiff-ABS, which denotes the method working by minimizing the corrected risk estimator with the absolute value function as the risk correction function. We compared our proposed approaches with several Pcomp methods [43], including Pcomp-Unbiased, Pcomp-ReLU, Pcomp-ABS, and Pcomp-Teacher. We also recorded the experimental results of supervised learning methods, including Oracle-Hard having access to ground-truth hard labels and Oracle-Soft having access to pointwise positive confidence. All the experiments were conducted on NVIDIA GeForce RTX 3090. The number of training epochs was set to 200 and we obtained the testing accuracy by averaging the results in the last 10 epochs. All the methods were implemented in PyTorch [69]. We used the Adam optimizer [70]. To ensure fair comparisons, We set the same hyperparameter values for all the compared approaches, where the details can be found in Appendix H.

## 4.2 Experimental Results

**Benchmark data sets.** Table 1 reports detailed experimental results for all the compared methods on four benchmark data sets. Based on Table 1, we can draw the following conclusions: a) On all the cases of benchmark data sets, our proposed ConfDiff-ABS method achieves superior performance against all of the other compared approaches significantly, which validates the effectiveness of our approach in utilizing supervision information of confidence difference; b) Pcomp-Teacher achieves

Table 2: Classification accuracy (mean±std) of each method on UCI data sets with different class priors, where the best performance (excluding Oracle) is shown in bold.

| Class Prior | Method | Optdigits | USPS | Pendigits | Letter |
|---|---|---|---|---|---|
| $\pi_+ = 0.2$ | Pcomp-Unbiased | 0.771±0.016 | 0.721±0.046 | 0.743±0.057 | 0.757±0.028 |
| | Pcomp-ReLU | 0.800±0.000 | 0.800±0.000 | 0.800±0.000 | 0.800±0.000 |
| | Pcomp-ABS | 0.800±0.001 | 0.800±0.000 | 0.800±0.000 | 0.800±0.000 |
| | Pcomp-Teacher | 0.901±0.023 | 0.894±0.023 | 0.928±0.019 | 0.883±0.006 |
| | Oracle-Hard | 0.990±0.002 | 0.984±0.002 | 0.997±0.001 | 0.978±0.003 |
| | Oracle-Soft | 0.990±0.003 | 0.984±0.004 | 0.998±0.001 | 0.971±0.007 |
| | ConfDiff-Unbiased | 0.831±0.078 | 0.840±0.078 | 0.865±0.079 | 0.732±0.053 |
| | ConfDiff-ReLU | 0.953±0.014 | 0.957±0.007 | 0.987±0.003 | 0.929±0.008 |
| | ConfDiff-ABS | **0.963±0.009** | **0.960±0.005** | **0.988±0.002** | **0.942±0.007** |
| $\pi_+ = 0.5$ | Pcomp-Unbiased | 0.651±0.112 | 0.671±0.090 | 0.748±0.038 | 0.632±0.019 |
| | Pcomp-ReLU | 0.630±0.076 | 0.554±0.048 | 0.514±0.019 | 0.525±0.023 |
| | Pcomp-ABS | 0.787±0.031 | 0.814±0.018 | 0.793±0.017 | 0.748±0.031 |
| | Pcomp-Teacher | 0.890±0.009 | 0.860±0.012 | 0.883±0.018 | 0.864±0.024 |
| | Oracle-Hard | 0.988±0.003 | 0.980±0.003 | 0.997±0.001 | 0.975±0.001 |
| | Oracle-Soft | 0.987±0.003 | 0.980±0.003 | 0.997±0.001 | 0.967±0.006 |
| | ConfDiff-Unbiased | 0.917±0.006 | 0.936±0.010 | 0.945±0.052 | 0.755±0.041 |
| | ConfDiff-ReLU | 0.921±0.011 | 0.945±0.009 | 0.981±0.004 | 0.895±0.006 |
| | ConfDiff-ABS | **0.962±0.006** | **0.959±0.004** | **0.988±0.003** | **0.925±0.003** |
| $\pi_+ = 0.8$ | Pcomp-Unbiased | 0.765±0.023 | 0.746±0.012 | 0.743±0.026 | 0.694±0.031 |
| | Pcomp-ReLU | 0.902±0.017 | 0.891±0.024 | 0.913±0.023 | 0.827±0.025 |
| | Pcomp-ABS | 0.894±0.019 | 0.879±0.009 | 0.911±0.009 | 0.870±0.006 |
| | Pcomp-Teacher | 0.918±0.007 | 0.933±0.023 | 0.903±0.008 | 0.872±0.011 |
| | Oracle-Hard | 0.987±0.003 | 0.983±0.002 | 0.997±0.001 | 0.976±0.004 |
| | Oracle-Soft | 0.986±0.003 | 0.985±0.004 | 0.998±0.001 | 0.965±0.010 |
| | ConfDiff-Unbiased | 0.886±0.037 | 0.803±0.042 | 0.892±0.096 | 0.748±0.015 |
| | ConfDiff-ReLU | 0.949±0.007 | 0.958±0.008 | 0.986±0.003 | 0.927±0.008 |
| | ConfDiff-ABS | **0.964±0.005** | **0.964±0.003** | **0.987±0.002** | **0.945±0.007** |

superior performance against all of the other Pcomp approaches by a large margin. The excellent performance benefits from the effectiveness of consistency regularization for weakly supervised learning problems [1, 17, 71]; c) It is worth noting that the classification results of ConfDiff-ReLU and ConfDiff-ABS have smaller variances than ConfDiff-Unbiased. It demonstrates that the risk correction method can enhance the stability and robustness for ConfDiff classification.

**UCI data sets.** Table 2 reports detailed experimental results on four UCI data sets as well. From Table 2, we can observe that: a) On all the UCI data sets under different class prior probability settings, our proposed ConfDiff-ABS method achieves the best performance among all the compared approaches with significant superiority, which verifies the effectiveness of our proposed approaches again; b) The performance of our proposed approaches is more stable than the compared Pcomp approaches under different class prior probability settings, demonstrating the superiority of our methods in dealing with various kinds of data distributions.

## 4.3 Experiments on a Real-world Recommender System Data Set

We also conducted experiments on a recommender system data set to demonstrate our approach's usefulness and promising applications in real-world scenarios. We used the KuaiRec [72] data set, a real-world recommender system data set collected from a well-known short-video mobile app. In this data set, user-item interactions are represented by watching ratios, i.e., the ratios of watching time to the entire length of videos. Such statistics could reveal the confidence of preference, and we regarded them as pointwise positive confidence. We generated pairwise confidence difference between pairs of items for a given user. We adopted the NCF [73] model as our backbone.

Table 3: Experimental results on the KuaiRec data set.

| Method | HR | NDCG |
|---|---|---|
| BPR | 0.464 | 0.256 |
| MRL | 0.476 | 0.271 |
| Oracle-Hard | 0.469 | 0.283 |
| Oracle-Soft | 0.534 | 0.380 |
| Pcomp-Teacher | 0.179 | 0.066 |
| ConfDiff-ABS | 0.570 | 0.372 |

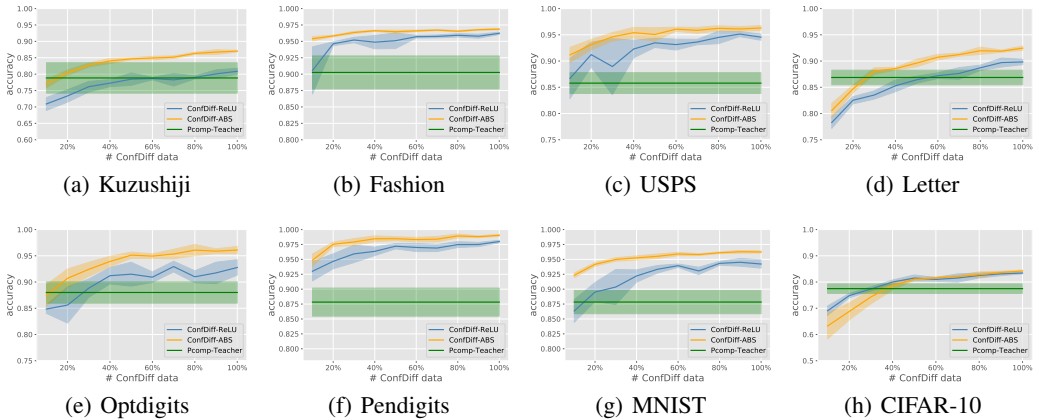

Figure 2: Classification performance of ConfDiff-ReLU and ConfDiff-ABS given a fraction of training data as well as Pcomp-Teacher given 100% of training data ($\pi_+ = 0.5$).

Details of the experimental setup and the data set can be found in Appendix H. We employed 5 compared methods, including BPR [74], Margin Ranking Loss (MRL), Oracle-Hard, Oracle-Soft, and Pcomp-Teacher. Table 3 reports the hit ratio (HR) and normalized discounted cumulative gain (NDCG) results. Our approach performs comparably against Oracle in terms of NDCG and even performs better in terms of HR. On the contrary, Pcomp-Teacher does not perform well on this data set. It validates the effectiveness of our approach in exploiting the supervision information of the confidence difference.

### 4.4 Performance with Fewer Training Data

We conducted experiments by changing the fraction of training data for ConfDiff-ReLU and ConfDiff-ABS (100% indicated that all the ConfDiff data were used for training). For comparison, we used 100% of training data for Pcomp-Teacher during the training process. Figure 2 shows the results with $\pi_+ = 0.5$, and more experimental results can be found in Appendix I. We can observe that the classification performance of our proposed approaches is still advantageous given a fraction of training data. Our approaches can achieve superior or comparable performance even when only 10% of training data are used. It elucidates that leveraging confidence difference may be more effective than increasing the number of training examples.

### 4.5 Analysis on Robustness

In this subsection, we investigate the influence of an inaccurate class prior probability and noisy confidence difference on the generalization performance of the proposed approaches. Specifically, let $\bar{\pi}_+ = \epsilon \pi_+$ denote the corrupted class prior probability with $\epsilon$ being a real number around 1. Let $\bar{c}_i = \epsilon'_i c_i$ denote the noisy confidence difference where $\epsilon'_i$ is sampled from a normal distribution $\mathcal{N}(1, \sigma^2)$. Figure 3 shows the classification performance of our proposed approaches on MNIST and Pendigits ($\pi_+ = 0.5$) with different $\epsilon$ and $\sigma$. It is demonstrated that the performance degenerates with $\epsilon = 0.8$ or $\epsilon = 1.2$ on some data sets, which indicates that it is important to estimate the class prior accurately.

## 5 Conclusion

In this paper, we dived into a novel weakly supervised learning setting where only unlabeled data pairs equipped with confidence difference were given. To solve the problem, an unbiased risk estimator was derived to perform empirical risk minimization. An estimation error bound was established to show that the optimal parametric convergence rate can be achieved. Furthermore, a risk correction approach was introduced to alleviate overfitting issues. Extensive experimental results validated the superiority of our proposed approaches. In future, it would be promising to apply our approaches in real-world scenarios and multi-class classification settings.

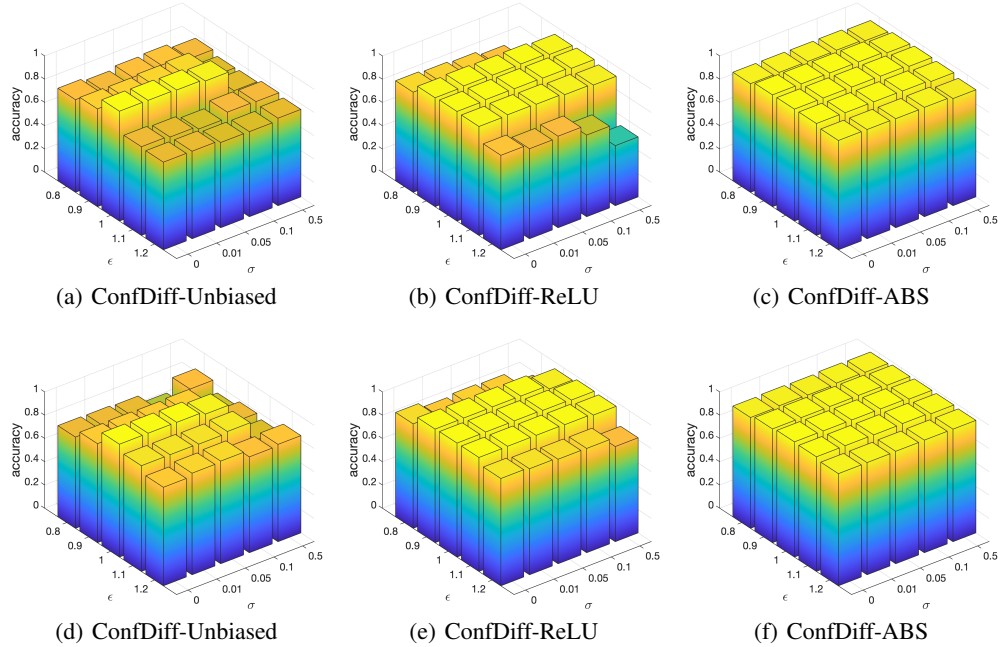

Figure 3: Classification accuracy on MNIST (the first row) and Pendigits (the second row) with $\pi_+ = 0.5$ given an inaccurate class prior probability and noisy confidence difference.

## Acknowledgments and Disclosure of Funding

The authors wish to thank the anonymous reviewers for their helpful comments and suggestions. The research was partially supported by the SGU MEXT Scholarship, the Junior Research Associate (JRA) program of RIKEN, Microsoft Research Asia, the National Science Foundation of China (62176055), and Basic Research Grant (Super AI) of Institute for AI and Beyond of the University of Tokyo.

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

# A  Proof of Theorem 1

Before giving the proof of Theorem 1, we begin with the following lemmas:

**Lemma 2.** *The confidence difference $c(\boldsymbol{x}, \boldsymbol{x}')$ can be equivalently expressed as*

$$c(\boldsymbol{x}, \boldsymbol{x}') = \frac{\pi_+ p(\boldsymbol{x}) p_+(\boldsymbol{x}') - \pi_+ p_+(\boldsymbol{x}) p(\boldsymbol{x}')}{p(\boldsymbol{x}) p(\boldsymbol{x}')} \tag{16}$$

$$= \frac{\pi_- p_-(\boldsymbol{x}) p(\boldsymbol{x}') - \pi_- p(\boldsymbol{x}) p_-(\boldsymbol{x}')}{p(\boldsymbol{x}) p(\boldsymbol{x}')} \tag{17}$$

*Proof.* On one hand,

$$
\begin{aligned}
c(\boldsymbol{x}, \boldsymbol{x}') &= p(y' = +1 | \boldsymbol{x}') - p(y = +1 | \boldsymbol{x}) \\
&= \frac{p(\boldsymbol{x}', y' = +1)}{p(\boldsymbol{x}')} - \frac{p(\boldsymbol{x}, y = +1)}{p(\boldsymbol{x})} \\
&= \frac{\pi_+ p_+(\boldsymbol{x}')}{p(\boldsymbol{x}')} - \frac{\pi_+ p_+(\boldsymbol{x})}{p(\boldsymbol{x})} \\
&= \frac{\pi_+ p(\boldsymbol{x}) p_+(\boldsymbol{x}') - \pi_+ p_+(\boldsymbol{x}) p(\boldsymbol{x}')}{p(\boldsymbol{x}) p(\boldsymbol{x}')}.
\end{aligned}
$$

On the other hand,

$$
\begin{aligned}
c(\boldsymbol{x}, \boldsymbol{x}') &= p(y' = +1 | \boldsymbol{x}') - p(y = +1 | \boldsymbol{x}) \\
&= (1 - p(y' = 0 | \boldsymbol{x}')) - (1 - p(y = 0 | \boldsymbol{x})) \\
&= p(y = 0 | \boldsymbol{x}) - p(y' = 0 | \boldsymbol{x}') \\
&= \frac{p(\boldsymbol{x}, y = 0)}{p(\boldsymbol{x})} - \frac{p(\boldsymbol{x}', y = 0)}{p(\boldsymbol{x}')} \\
&= \frac{\pi_- p_-(\boldsymbol{x})}{p(\boldsymbol{x})} - \frac{\pi_- p_-(\boldsymbol{x}')}{p(\boldsymbol{x}')} \\
&= \frac{\pi_- p_-(\boldsymbol{x}) p(\boldsymbol{x}') - \pi_- p(\boldsymbol{x}) p_-(\boldsymbol{x}')}{p(\boldsymbol{x}) p(\boldsymbol{x}')},
\end{aligned}
$$

which concludes the proof. $\square$

**Lemma 3.** *The following equations hold:*

$$\mathbb{E}_{p(\boldsymbol{x},\boldsymbol{x}')}[(\pi_+ - c(\boldsymbol{x}, \boldsymbol{x}'))\ell(g(\boldsymbol{x}), +1)] = \pi_+ \mathbb{E}_{p_+(\boldsymbol{x})}[\ell(g(\boldsymbol{x}), +1)], \tag{18}$$

$$\mathbb{E}_{p(\boldsymbol{x},\boldsymbol{x}')}[(\pi_- + c(\boldsymbol{x}, \boldsymbol{x}'))\ell(g(\boldsymbol{x}), -1)] = \pi_- \mathbb{E}_{p_-(\boldsymbol{x})}[\ell(g(\boldsymbol{x}), -1)], \tag{19}$$

$$\mathbb{E}_{p(\boldsymbol{x},\boldsymbol{x}')}[(\pi_+ + c(\boldsymbol{x}, \boldsymbol{x}'))\ell(g(\boldsymbol{x}'), +1)] = \pi_+ \mathbb{E}_{p_+(\boldsymbol{x}')}[\ell(g(\boldsymbol{x}'), +1)], \tag{20}$$

$$\mathbb{E}_{p(\boldsymbol{x},\boldsymbol{x}')}[(\pi_- - c(\boldsymbol{x}, \boldsymbol{x}'))\ell(g(\boldsymbol{x}'), -1)] = \pi_- \mathbb{E}_{p_-(\boldsymbol{x}')}[\ell(g(\boldsymbol{x}'), -1)]. \tag{21}$$

*Proof.* Firstly, the proof of Eq. (18) is given:

$$\mathbb{E}_{p(\boldsymbol{x},\boldsymbol{x}')}[(\pi_+ - c(\boldsymbol{x},\boldsymbol{x}'))\ell(g(\boldsymbol{x}),+1)]$$

$$= \int\int \frac{\pi_+ p(\boldsymbol{x})p(\boldsymbol{x}') - \pi_+ p(\boldsymbol{x})p_+(\boldsymbol{x}') + \pi_+ p_+(\boldsymbol{x})p(\boldsymbol{x}')}{p(\boldsymbol{x})p(\boldsymbol{x}')}\ell(g(\boldsymbol{x}),+1)p(\boldsymbol{x},\boldsymbol{x}')\,\mathrm{d}\boldsymbol{x}\,\mathrm{d}\boldsymbol{x}'$$

$$= \int\int (\pi_+ p(\boldsymbol{x})p(\boldsymbol{x}') - \pi_+ p(\boldsymbol{x})p_+(\boldsymbol{x}') + \pi_+ p_+(\boldsymbol{x})p(\boldsymbol{x}'))\ell(g(\boldsymbol{x}),+1)\,\mathrm{d}\boldsymbol{x}\,\mathrm{d}\boldsymbol{x}'$$

$$= \int \pi_+ p(\boldsymbol{x})\ell(g(\boldsymbol{x}),+1)\,\mathrm{d}\boldsymbol{x}\int p(\boldsymbol{x}')\,\mathrm{d}\boldsymbol{x}' - \int \pi_+ p(\boldsymbol{x})\ell(g(\boldsymbol{x}),+1)\,\mathrm{d}\boldsymbol{x}\int p_+(\boldsymbol{x}')\,\mathrm{d}\boldsymbol{x}'$$

$$+ \int \pi_+ p_+(\boldsymbol{x})\ell(g(\boldsymbol{x}),+1)\,\mathrm{d}\boldsymbol{x}\int p(\boldsymbol{x}')\,\mathrm{d}\boldsymbol{x}'$$

$$= \int \pi_+ p(\boldsymbol{x})\ell(g(\boldsymbol{x}),+1)\,\mathrm{d}\boldsymbol{x} - \int \pi_+ p(\boldsymbol{x})\ell(g(\boldsymbol{x}),+1)\,\mathrm{d}\boldsymbol{x} + \int \pi_+ p_+(\boldsymbol{x})\ell(g(\boldsymbol{x}),+1)\,\mathrm{d}\boldsymbol{x}$$

$$= \int \pi_+ p_+(\boldsymbol{x})\ell(g(\boldsymbol{x}),+1)\,\mathrm{d}\boldsymbol{x}$$

$$= \pi_+ \mathbb{E}_{p_+(\boldsymbol{x})}[\ell(g(\boldsymbol{x}),+1)].$$

After that, the proof of Eq. (19) is given:

$$\mathbb{E}_{p(\boldsymbol{x},\boldsymbol{x}')}[(\pi_- + c(\boldsymbol{x},\boldsymbol{x}'))\ell(g(\boldsymbol{x}),-1)]$$

$$= \int\int \frac{\pi_- p(\boldsymbol{x})p(\boldsymbol{x}') + \pi_- p_-(\boldsymbol{x})p(\boldsymbol{x}') - \pi_- p(\boldsymbol{x})p_-(\boldsymbol{x}')}{p(\boldsymbol{x})p(\boldsymbol{x}')}\ell(g(\boldsymbol{x}),-1)p(\boldsymbol{x},\boldsymbol{x}')\,\mathrm{d}\boldsymbol{x}\,\mathrm{d}\boldsymbol{x}'$$

$$= \int\int (\pi_- p(\boldsymbol{x})p(\boldsymbol{x}') + \pi_- p_-(\boldsymbol{x})p(\boldsymbol{x}') - \pi_- p(\boldsymbol{x})p_-(\boldsymbol{x}'))\ell(g(\boldsymbol{x}),-1)\,\mathrm{d}\boldsymbol{x}\,\mathrm{d}\boldsymbol{x}'$$

$$= \int \pi_- p(\boldsymbol{x})\ell(g(\boldsymbol{x}),-1)\,\mathrm{d}\boldsymbol{x}\int p(\boldsymbol{x}')\,\mathrm{d}\boldsymbol{x}' + \int \pi_- p_-(\boldsymbol{x})\ell(g(\boldsymbol{x}),-1)\,\mathrm{d}\boldsymbol{x}\int p(\boldsymbol{x}')\,\mathrm{d}\boldsymbol{x}'$$

$$- \int \pi_- p(\boldsymbol{x})\ell(g(\boldsymbol{x}),-1)\,\mathrm{d}\boldsymbol{x}\int p_-(\boldsymbol{x}')\,\mathrm{d}\boldsymbol{x}'$$

$$= \int \pi_- p(\boldsymbol{x})\ell(g(\boldsymbol{x}),-1)\,\mathrm{d}\boldsymbol{x} + \int \pi_- p_-(\boldsymbol{x})\ell(g(\boldsymbol{x}),-1)\,\mathrm{d}\boldsymbol{x} - \int \pi_- p(\boldsymbol{x})\ell(g(\boldsymbol{x}),-1)\,\mathrm{d}\boldsymbol{x}$$

$$= \int \pi_- p_-(\boldsymbol{x})\ell(g(\boldsymbol{x}),-1)\,\mathrm{d}\boldsymbol{x}$$

$$= \pi_- \mathbb{E}_{p_-(\boldsymbol{x})}[\ell(g(\boldsymbol{x}),-1)].$$

It can be noticed that $c(\boldsymbol{x},\boldsymbol{x}') = -c(\boldsymbol{x}',\boldsymbol{x})$ and $p(\boldsymbol{x},\boldsymbol{x}') = p(\boldsymbol{x}',\boldsymbol{x})$. Therefore, it can be deduced naturally that $\mathbb{E}_{p(\boldsymbol{x},\boldsymbol{x}')}[(\pi_+ - c(\boldsymbol{x},\boldsymbol{x}'))\ell(g(\boldsymbol{x}),+1)] = \mathbb{E}_{p(\boldsymbol{x}',\boldsymbol{x})}[(\pi_+ + c(\boldsymbol{x}',\boldsymbol{x}))\ell(g(\boldsymbol{x}),+1)]$. Because $\boldsymbol{x}$ and $\boldsymbol{x}'$ are symmetric, we can swap them and deduce Eq. (20). Eq. (21) can be deduced in the same manner, which concludes the proof. □

Based on Lemma 3, the proof of Theorem 1 is given.

*Proof of Theorem 1.* To begin with, it can be noticed that $\mathbb{E}_{p_+(\boldsymbol{x})}[\ell(g(\boldsymbol{x}),+1)] = \mathbb{E}_{p_+(\boldsymbol{x}')}[\ell(g(\boldsymbol{x}'),+1)]$ and $\mathbb{E}_{p_-(\boldsymbol{x})}[\ell(g(\boldsymbol{x}),-1)] = \mathbb{E}_{p_-(\boldsymbol{x}')}[\ell(g(\boldsymbol{x}'),-1)]$. Then, by summing up all the equations from Eq. (18) to Eq. (21), we can get the following equation:

$$\mathbb{E}_{p(\boldsymbol{x},\boldsymbol{x}')}[\mathcal{L}_+(g(\boldsymbol{x}),g(\boldsymbol{x}')) + \mathcal{L}_-(g(\boldsymbol{x}),g(\boldsymbol{x}'))]$$
$$= 2\pi_+ \mathbb{E}_{p_+(\boldsymbol{x})}[\ell(g(\boldsymbol{x}),+1)] + 2\pi_- \mathbb{E}_{p_-(\boldsymbol{x})}[\ell(g(\boldsymbol{x}),-1)]$$

After dividing each side of the equation above by 2, we can obtain Theorem 1. □

## B  Analysis on Variance of Risk Estimator

### B.1  Proof of Lemma 1

Based on Lemma 3, it can be observed that

$$
\begin{aligned}
\mathbb{E}_{p(\boldsymbol{x},\boldsymbol{x}')}[\mathcal{L}(\boldsymbol{x},\boldsymbol{x}')] &= \mathbb{E}_{p(\boldsymbol{x},\boldsymbol{x}')}[(\pi_+ - c(\boldsymbol{x},\boldsymbol{x}'))\ell(g(\boldsymbol{x}),+1) + (\pi_- - c(\boldsymbol{x},\boldsymbol{x}'))\ell(g(\boldsymbol{x}'),-1)] \\
&= \pi_+\mathbb{E}_{p_+(\boldsymbol{x})}[\ell(g(\boldsymbol{x}),+1)] + \pi_-\mathbb{E}_{p_-(\boldsymbol{x}')}[\ell(g(\boldsymbol{x}'),-1)] \\
&= \pi_+\mathbb{E}_{p_+(\boldsymbol{x})}[\ell(g(\boldsymbol{x}),+1)] + \pi_-\mathbb{E}_{p_-(\boldsymbol{x})}[\ell(g(\boldsymbol{x}),-1)] \\
&= R(g)
\end{aligned}
$$

and

$$
\begin{aligned}
\mathbb{E}_{p(\boldsymbol{x},\boldsymbol{x}')}[\mathcal{L}(\boldsymbol{x}',\boldsymbol{x})] &= \mathbb{E}_{p(\boldsymbol{x},\boldsymbol{x}')}[(\pi_+ + c(\boldsymbol{x},\boldsymbol{x}'))\ell(g(\boldsymbol{x}'),+1) + (\pi_- + c(\boldsymbol{x},\boldsymbol{x}'))\ell(g(\boldsymbol{x}),-1)] \\
&= \pi_-\mathbb{E}_{p_-(\boldsymbol{x})}[\ell(g(\boldsymbol{x}),-1)] + \pi_+\mathbb{E}_{p_+(\boldsymbol{x}')}[\ell(g(\boldsymbol{x}'),+1)] \\
&= \pi_-\mathbb{E}_{p_-(\boldsymbol{x})}[\ell(g(\boldsymbol{x}),-1)] + \pi_+\mathbb{E}_{p_+(\boldsymbol{x})}[\ell(g(\boldsymbol{x}),+1)] \\
&= R(g).
\end{aligned}
$$

Therefore, for an arbitrary weight $\alpha \in [0,1]$,

$$
\begin{aligned}
R(g) &= \alpha R(g) + (1-\alpha)R(g) \\
&= \alpha\mathbb{E}_{p(\boldsymbol{x},\boldsymbol{x}')}[\mathcal{L}(\boldsymbol{x},\boldsymbol{x}')] + (1-\alpha)\mathbb{E}_{p(\boldsymbol{x},\boldsymbol{x}')}[\mathcal{L}(\boldsymbol{x}',\boldsymbol{x})],
\end{aligned}
$$

which indicates that

$$
\frac{1}{n}\sum_{i=1}^{n}(\alpha\mathcal{L}(\boldsymbol{x}_i,\boldsymbol{x}'_i) + (1-\alpha)\mathcal{L}(\boldsymbol{x}'_i,\boldsymbol{x}_i))
$$

is also an unbiased risk estimator and concludes the proof.  □

### B.2  Proof of Theorem 2

In this subsection, we show that Eq. (8) in the main paper achieves the minimum variance of

$$
S(g;\alpha) = \frac{1}{n}\sum_{i=1}^{n}(\alpha\mathcal{L}(\boldsymbol{x}_i,\boldsymbol{x}'_i) + (1-\alpha)\mathcal{L}(\boldsymbol{x}'_i,\boldsymbol{x}_i))
$$

w.r.t. any $\alpha \in [0,1]$. To begin with, we introduce the following notations:

$$
\mu_1 \triangleq \mathbb{E}_{p(\boldsymbol{x},\boldsymbol{x}')}[(\frac{1}{n}\sum_{i=1}^{n}\mathcal{L}(\boldsymbol{x}_i,\boldsymbol{x}'_i))^2] = \mathbb{E}_{p(\boldsymbol{x},\boldsymbol{x}')}[(\frac{1}{n}\sum_{i=1}^{n}\mathcal{L}(\boldsymbol{x}'_i,\boldsymbol{x}_i))^2],
$$

$$
\mu_2 \triangleq \mathbb{E}_{p(\boldsymbol{x},\boldsymbol{x}')}[\frac{1}{n^2}\sum_{i=1}^{n}\mathcal{L}(\boldsymbol{x}_i,\boldsymbol{x}'_i)\sum_{i=1}^{n}\mathcal{L}(\boldsymbol{x}'_i,\boldsymbol{x}_i)].
$$

Furthermore, according to Lemma 1 in the main paper, we have

$$
\mathbb{E}_{p(\boldsymbol{x},\boldsymbol{x}')}[S(g;\alpha)] = R(g).
$$

Then, we provide the proof of Theorem 2 as follows.

*Proof of Theorem 2.*

$$
\begin{aligned}
\mathrm{Var}(S(g;\alpha)) &= \mathbb{E}_{p(\boldsymbol{x},\boldsymbol{x}')}[(S(g;\alpha) - R(g))^2] \\
&= \mathbb{E}_{p(\boldsymbol{x},\boldsymbol{x}')}[S(g;\alpha)^2] - R(g)^2 \\
&= \alpha^2\mathbb{E}_{p(\boldsymbol{x},\boldsymbol{x}')}[(\frac{1}{n}\sum_{i=1}^{n}\mathcal{L}(\boldsymbol{x}_i,\boldsymbol{x}'_i))^2] + (1-\alpha)^2\mathbb{E}_{p(\boldsymbol{x},\boldsymbol{x}')}[(\frac{1}{n}\sum_{i=1}^{n}\mathcal{L}(\boldsymbol{x}'_i,\boldsymbol{x}_i))^2] \\
&\quad + 2\alpha(1-\alpha)\mathbb{E}_{p(\boldsymbol{x},\boldsymbol{x}')}[\frac{1}{n^2}\sum_{i=1}^{n}\mathcal{L}(\boldsymbol{x}_i,\boldsymbol{x}'_i)\sum_{i=1}^{n}\mathcal{L}(\boldsymbol{x}'_i,\boldsymbol{x}_i)] - R(g)^2 \\
&= \mu_1\alpha^2 + \mu_1(1-\alpha)^2 + 2\mu_2\alpha(1-\alpha) - R(g)^2 \\
&= (2\mu_1 - 2\mu_2)(\alpha - \frac{1}{2})^2 + \frac{1}{2}(\mu_1 + \mu_2) - R(g)^2.
\end{aligned}
$$

Besides, it can be observed that

$$2\mu_1 - 2\mu_2 = \mathbb{E}_{p(\boldsymbol{x}, \boldsymbol{x}')}[(\frac{1}{n}\sum_{i=1}^{n}(\mathcal{L}(\boldsymbol{x}_i, \boldsymbol{x}'_i) - \mathcal{L}(\boldsymbol{x}'_i, \boldsymbol{x}_i)))^2] \geq 0.$$

Therefore, $\mathrm{Var}(S(g; \alpha))$ achieves the minimum value when $\alpha = 1/2$, which concludes the proof. $\square$

## C  Proof of Theorem 3

To begin with, we give the definition of Rademacher complexity.

**Definition 2** (Rademacher complexity). *Let $\mathcal{X}_n = \{\boldsymbol{x}_1, \cdots \boldsymbol{x}_n\}$ denote $n$ i.i.d. random variables drawn from a probability distribution with density $p(\boldsymbol{x})$, $\mathcal{G} = \{g : \mathcal{X} \mapsto \mathbb{R}\}$ denote a class of measurable functions, and $\boldsymbol{\sigma} = (\sigma_1, \sigma_2, \cdots, \sigma_n)$ denote Rademacher variables taking values from $\{+1, -1\}$ uniformly. Then, the (expected) Rademacher complexity of $\mathcal{G}$ is defined as*

$$\mathfrak{R}_n(\mathcal{G}) = \mathbb{E}_{\mathcal{X}_n}\mathbb{E}_{\boldsymbol{\sigma}}\left[\sup_{g \in \mathcal{G}}\frac{1}{n}\sum_{i=1}^{n}\sigma_i g(\boldsymbol{x}_i)\right]. \tag{22}$$

Let $\mathcal{D}_n \overset{\text{i.i.d.}}{\sim} p(\boldsymbol{x}, \boldsymbol{x}')$ denote $n$ pairs of ConfDiff data and $\mathcal{L}_{\mathrm{CD}}(g; \boldsymbol{x}_i, \boldsymbol{x}'_i) = (\mathcal{L}(\boldsymbol{x}, \boldsymbol{x}') + \mathcal{L}(\boldsymbol{x}', \boldsymbol{x}))/2$, then we introduce the following lemma.

**Lemma 4.**

$$\bar{\mathfrak{R}}_n(\mathcal{L}_{\mathrm{CD}} \circ \mathcal{G}) \leq 2L_\ell \mathfrak{R}_n(\mathcal{G}),$$

*where $\mathcal{L}_{\mathrm{CD}} \circ \mathcal{G} = \{\mathcal{L}_{\mathrm{CD}} \circ g | g \in \mathcal{G}\}$ and $\bar{\mathfrak{R}}_n(\cdot)$ is the Rademacher complexity over ConfDiff data pairs $\mathcal{D}_n$ of size $n$.*

*Proof.*

$$\begin{aligned}
\bar{\mathfrak{R}}_n(\mathcal{L}_{\mathrm{CD}} \circ \mathcal{G}) =& \mathbb{E}_{\mathcal{D}_n}\mathbb{E}_{\boldsymbol{\sigma}}[\sup_{g \in \mathcal{G}}\frac{1}{n}\sum_{i=1}^{n}\sigma_i \mathcal{L}_{\mathrm{CD}}(g; \boldsymbol{x}_i, \boldsymbol{x}'_i)] \\
=& \mathbb{E}_{\mathcal{D}_n}\mathbb{E}_{\boldsymbol{\sigma}}[\sup_{g \in \mathcal{G}}\frac{1}{2n}\sum_{i=1}^{n}\sigma_i((\pi_+ - c_i)\ell(g(\boldsymbol{x}_i), +1) + (\pi_- - c_i)\ell(g(\boldsymbol{x}'_i), -1) \\
& + (\pi_+ + c_i)\ell(g(\boldsymbol{x}'_i), +1) + (\pi_- + c_i)\ell(g(\boldsymbol{x}_i), -1))].
\end{aligned}$$

Then, we can induce that

$$\begin{aligned}
& \|\nabla \mathcal{L}_{\mathrm{CD}}(g; \boldsymbol{x}_i, \boldsymbol{x}'_i)\|_2 \\
=& \|\nabla(\frac{(\pi_+ - c_i)\ell(g(\boldsymbol{x}_i), +1) + (\pi_- - c_i)\ell(g(\boldsymbol{x}'_i), -1)}{2} \\
& + \frac{(\pi_+ + c_i)\ell(g(\boldsymbol{x}'_i), +1) + (\pi_- + c_i)\ell(g(\boldsymbol{x}_i), -1)}{2})\|_2 \\
\leq& \|\nabla(\frac{(\pi_+ - c_i)\ell(g(\boldsymbol{x}_i), +1)}{2})\|_2 + \|\nabla(\frac{(\pi_- - c_i)\ell(g(\boldsymbol{x}'_i), -1)}{2})\|_2 \\
& + \|\nabla(\frac{(\pi_+ + c_i)\ell(g(\boldsymbol{x}'_i), +1)}{2})\|_2 + \|\nabla(\frac{(\pi_- + c_i)\ell(g(\boldsymbol{x}_i), -1)}{2})\|_2 \\
\leq& \frac{|\pi_+ - c_i|L_\ell}{2} + \frac{|\pi_- - c_i|L_\ell}{2} + \frac{|\pi_+ + c_i|L_\ell}{2} + \frac{|\pi_- + c_i|L_\ell}{2}. \tag{23}
\end{aligned}$$

Suppose $\pi_+ \geq \pi_-$, the value of RHS of Eq. (23) can be determined as follows: when $c_i \in [-1, -\pi_+)$, the value is $-2c_i L_\ell$; when $c_i \in [-\pi_+, -\pi_-)$, the value is $(\pi_+ - c_i)L_\ell$; when $c_i \in [-\pi_-, \pi_-)$, the value is $L_\ell$; when $c_i \in [\pi_-, \pi_+)$, the value is $(\pi_+ + c_i)L_\ell$; when $c_i \in [\pi_+, 1]$, the value is $2c_i L_\ell$. To sum up, when $\pi_+ \geq \pi_-$, the value of RHS of Eq. (23) is less than $2L_\ell$. When $\pi_+ \leq \pi_-$, we can

deduce that the value of RHS of Eq. (23) is less than $2L_\ell$ in the same way. Therefore,

$$\bar{\mathfrak{R}}_n(\mathcal{L}_{\mathrm{CD}} \circ \mathcal{G}) \leq 2L_\ell \mathbb{E}_{\mathcal{D}_n} \mathbb{E}_{\boldsymbol{\sigma}} [\sup_{g \in \mathcal{G}} \frac{1}{n} \sum_{i=1}^n \sigma_i g(\boldsymbol{x}_i)]$$

$$= 2L_\ell \mathbb{E}_{\mathcal{X}_n} \mathbb{E}_{\boldsymbol{\sigma}} [\sup_{g \in \mathcal{G}} \frac{1}{n} \sum_{i=1}^n \sigma_i g(\boldsymbol{x}_i)]$$

$$= 2L_\ell \mathfrak{R}_n(\mathcal{G}),$$

which concludes the proof. $\qquad\square$

After that, we introduce the following lemma.

**Lemma 5.** *The inequality below hold with probability at least $1 - \delta$:*

$$\sup_{g \in \mathcal{G}} |R(g) - \widehat{R}_{\mathrm{CD}}(g)| \leq 4L_\ell \mathfrak{R}_n(\mathcal{G}) + 2C_\ell \sqrt{\frac{\ln 2/\delta}{2n}}.$$

*Proof.* To begin with, we introduce $\Phi = \sup_{g \in \mathcal{G}} (R(g) - \widehat{R}_{\mathrm{CD}}(g))$ and $\bar{\Phi} = \sup_{g \in \mathcal{G}} (R(g) - \widehat{\bar{R}}_{\mathrm{CD}}(g))$, where $\widehat{R}_{\mathrm{CD}}(g)$ and $\widehat{\bar{R}}_{\mathrm{CD}}(g)$ denote the empirical risk over two sets of training examples with exactly one different point $\{(\boldsymbol{x}_i, \boldsymbol{x}_i'), c_i\}$ and $\{(\bar{\boldsymbol{x}}_i, \bar{\boldsymbol{x}}_i'), c(\bar{\boldsymbol{x}}_i, \bar{\boldsymbol{x}}_i')\}$ respectively. Then we have

$$\bar{\Phi} - \Phi \leq \sup_{g \in \mathcal{G}} (\widehat{R}_{\mathrm{CD}}(g) - \widehat{\bar{R}}_{\mathrm{CD}}(g))$$

$$\leq \sup_{g \in \mathcal{G}} (\frac{\mathcal{L}_{\mathrm{CD}}(g; \boldsymbol{x}_i, \boldsymbol{x}_i') - \mathcal{L}_{\mathrm{CD}}(g; \bar{\boldsymbol{x}}_i, \bar{\boldsymbol{x}}_i')}{n})$$

$$\leq \frac{2C_\ell}{n}.$$

Accordingly, $\Phi - \bar{\Phi}$ can be bounded in the same way. The following inequalities holds with probability at least $1 - \delta/2$ by applying McDiarmid's inequality:

$$\sup_{g \in \mathcal{G}} (R(g) - \widehat{R}_{\mathrm{CD}}(g)) \leq \mathbb{E}_{\mathcal{D}_n} [\sup_{g \in \mathcal{G}} (R(g) - \widehat{R}_{\mathrm{CD}}(g))] + 2C_\ell \sqrt{\frac{\ln 2/\delta}{2n}},$$

Furthermore, we can bound $\mathbb{E}_{\mathcal{D}_n} [\sup_{g \in \mathcal{G}} (R(g) - \widehat{R}_{\mathrm{CD}}(g))]$ with Rademacher complexity. It is a routine work to show by symmetrization [60] that

$$\mathbb{E}_{\mathcal{D}_n} [\sup_{g \in \mathcal{G}} (R(g) - \widehat{R}_{\mathrm{CD}}(g))] \leq 2\bar{\mathfrak{R}}_n(\mathcal{L}_{\mathrm{CD}} \circ \mathcal{G}) \leq 4L_\ell \mathfrak{R}_n(\mathcal{G}),$$

where the second inequality is from Lemma 4. Accordingly, $\sup_{g \in \mathcal{G}} (\widehat{R}_{\mathrm{CD}}(g) - R(g))$ has the same bound. By using the union bound, the following inequality holds with probability at least $1 - \delta$:

$$\sup_{g \in \mathcal{G}} |R(g) - \widehat{R}_{\mathrm{CD}}(g)| \leq 4L_\ell \mathfrak{R}_n(\mathcal{G}) + 2C_\ell \sqrt{\frac{\ln 2/\delta}{2n}},$$

which concludes the proof. $\qquad\square$

Finally, the proof of Theorem 3 is provided.

*Proof of Theorem 3.*

$$R(\widehat{g}_{\mathrm{CD}}) - R(g^*) = (R(\widehat{g}_{\mathrm{CD}}) - \widehat{R}_{\mathrm{CD}}(\widehat{g}_{\mathrm{CD}})) + (\widehat{R}_{\mathrm{CD}}(\widehat{g}_{\mathrm{CD}}) - \widehat{R}_{\mathrm{CD}}(g^*)) + (\widehat{R}_{\mathrm{CD}}(g^*) - R(g^*))$$

$$\leq (R(\widehat{g}_{\mathrm{CD}}) - \widehat{R}_{\mathrm{CD}}(\widehat{g}_{\mathrm{CD}})) + (\widehat{R}_{\mathrm{CD}}(g^*) - R(g^*))$$

$$\leq |R(\widehat{g}_{\mathrm{CD}}) - \widehat{R}_{\mathrm{CD}}(\widehat{g}_{\mathrm{CD}})| + \left| \widehat{R}_{\mathrm{CD}}(g^*) - R(g^*) \right|$$

$$\leq 2 \sup_{g \in \mathcal{G}} |R(g) - \widehat{R}_{\mathrm{CD}}(g)|$$

$$\leq 8L_\ell \mathfrak{R}_n(\mathcal{G}) + 4C_\ell \sqrt{\frac{\ln 2/\delta}{2n}}.$$

The first inequality is derived because $\widehat{g}_{\mathrm{CD}}$ is the minimizer of $\widehat{R}_{\mathrm{CD}}(g)$. The last inequality is derived according to Lemma 5, which concludes the proof. $\qquad\square$

## D   Proof of Theorem 4

To begin with, we provide the following inequality:

$$
\begin{aligned}
&|\bar{R}_{\mathrm{CD}}(g) - \widehat{R}_{\mathrm{CD}}(g)| \\
=&\frac{1}{2n}|\sum_{i=1}^{n}((\bar{\pi}_+ - \pi_+ + c_i - \bar{c}_i)\ell(g(\boldsymbol{x}_i),+1) + (\bar{\pi}_- - \pi_- + c_i - \bar{c}_i)\ell(g(\boldsymbol{x}'_i),-1) \\
&+ (\bar{\pi}_+ - \pi_+ + \bar{c}_i - c_i)\ell(g(\boldsymbol{x}'_i),+1) + (\bar{\pi}_- - \pi_- + \bar{c}_i - c_i)\ell(g(\boldsymbol{x}_i),-1))| \\
\leq&\frac{1}{2n}\sum_{i=1}^{n}(|(\bar{\pi}_+ - \pi_+ + c_i - \bar{c}_i)\ell(g(\boldsymbol{x}_i),+1)| + |(\bar{\pi}_- - \pi_- + c_i - \bar{c}_i)\ell(g(\boldsymbol{x}'_i),-1)| \\
&+ |(\bar{\pi}_+ - \pi_+ + \bar{c}_i - c_i)\ell(g(\boldsymbol{x}'_i),+1)| + |(\bar{\pi}_- - \pi_- + \bar{c}_i - c_i)\ell(g(\boldsymbol{x}_i),-1)|) \\
=&\frac{1}{2n}\sum_{i=1}^{n}(|\bar{\pi}_+ - \pi_+ + c_i - \bar{c}_i|\ell(g(\boldsymbol{x}_i),+1) + |\bar{\pi}_- - \pi_- + c_i - \bar{c}_i|\ell(g(\boldsymbol{x}'_i),-1) \\
&+ |\bar{\pi}_+ - \pi_+ + \bar{c}_i - c_i|\ell(g(\boldsymbol{x}'_i),+1) + |\bar{\pi}_- - \pi_- + \bar{c}_i - c_i|\ell(g(\boldsymbol{x}_i),-1)) \\
\leq&\frac{1}{2n}\sum_{i=1}^{n}((|\bar{\pi}_+ - \pi_+| + |c_i - \bar{c}_i|)\ell(g(\boldsymbol{x}_i),+1) + (|\bar{\pi}_- - \pi_-| + |c_i - \bar{c}_i|)\ell(g(\boldsymbol{x}'_i),-1) \\
&+ (|\bar{\pi}_+ - \pi_+| + |\bar{c}_i - c_i|)\ell(g(\boldsymbol{x}'_i),+1) + (|\bar{\pi}_- - \pi_-| + |\bar{c}_i - c_i|)\ell(g(\boldsymbol{x}_i),-1)) \\
=&\frac{1}{2n}\sum_{i=1}^{n}((|\bar{\pi}_+ - \pi_+| + |c_i - \bar{c}_i|)\ell(g(\boldsymbol{x}_i),+1) + (|\pi_+ - \bar{\pi}_+| + |c_i - \bar{c}_i|)\ell(g(\boldsymbol{x}'_i),-1) \\
&+ (|\bar{\pi}_+ - \pi_+| + |\bar{c}_i - c_i|)\ell(g(\boldsymbol{x}'_i),+1) + (|\pi_+ - \bar{\pi}_+| + |\bar{c}_i - c_i|)\ell(g(\boldsymbol{x}_i),-1)) \\
\leq&\frac{2C_\ell \sum_{i=1}^{n}|\bar{c}_i - c_i|}{n} + 2C_\ell|\bar{\pi}_+ - \pi_+|.
\end{aligned}
$$

Then, we deduce the following inequality:

$$
\begin{aligned}
R(\bar{g}_{\mathrm{CD}}) - R(g^*) =& (R(\bar{g}_{\mathrm{CD}}) - \widehat{R}_{\mathrm{CD}}(\bar{g}_{\mathrm{CD}})) + (\widehat{R}_{\mathrm{CD}}(\bar{g}_{\mathrm{CD}}) - \bar{R}_{\mathrm{CD}}(\bar{g}_{\mathrm{CD}})) + (\bar{R}_{\mathrm{CD}}(\bar{g}_{\mathrm{CD}}) - \bar{R}_{\mathrm{CD}}(\widehat{g}_{\mathrm{CD}})) \\
&+ (\bar{R}_{\mathrm{CD}}(\widehat{g}_{\mathrm{CD}}) - \widehat{R}_{\mathrm{CD}}(\widehat{g}_{\mathrm{CD}})) + (\widehat{R}_{\mathrm{CD}}(\widehat{g}_{\mathrm{CD}}) - R(\widehat{g}_{\mathrm{CD}})) + (R(\widehat{g}_{\mathrm{CD}}) - R(g^*)) \\
\leq& 2\sup_{g\in\mathcal{G}}|R(g) - \widehat{R}_{\mathrm{CD}}(g)| + 2\sup_{g\in\mathcal{G}}|\bar{R}_{\mathrm{CD}}(g) - \widehat{R}_{\mathrm{CD}}(g)| + (R(\widehat{g}_{\mathrm{CD}}) - R(g^*)) \\
\leq& 4\sup_{g\in\mathcal{G}}|R(g) - \widehat{R}_{\mathrm{CD}}(g)| + 2\sup_{g\in\mathcal{G}}|\bar{R}_{\mathrm{CD}}(g) - \widehat{R}_{\mathrm{CD}}(g)| \\
\leq& 16L_\ell\mathfrak{R}_n(\mathcal{G}) + 8C_\ell\sqrt{\frac{\ln 2/\delta}{2n}} + \frac{4C_\ell \sum_{i=1}^{n}|\bar{c}_i - c_i|}{n} + 4C_\ell|\bar{\pi}_+ - \pi_+|.
\end{aligned}
$$

The first inequality is derived because $\bar{g}_{\mathrm{CD}}$ is the minimizer of $\bar{R}(g)$. The second and third inequality are derived according to the proof of Theorem 3 and Lemma 5 respectively. $\qquad\square$

## E   Proof of Theorem 5

To begin with, let $\mathfrak{D}_n^+(g) = \{\mathcal{D}_n|\widehat{A}(g) \geq 0 \cap \widehat{B}(g) \geq 0 \cap \widehat{C}(g) \geq 0 \cap \widehat{D}(g) \geq 0\}$ and $\mathfrak{D}_n^-(g) = \{\mathcal{D}_n|\widehat{A}(g) \leq 0 \cup \widehat{B}(g) \leq 0 \cup \widehat{C}(g) \leq 0 \cup \widehat{D}(g) \leq 0\}$. Before giving the proof of Theorem 5, we give the following lemma based on the assumptions in Section 3.

**Lemma 6.** *The probability measure of $\mathfrak{D}_n^-(g)$ can be bounded as follows:*

$$
\mathbb{P}(\mathfrak{D}_n^-(g)) \leq \exp\left(\frac{-2a^2 n}{C_\ell^2}\right) + \exp\left(\frac{-2b^2 n}{C_\ell^2}\right) + \exp\left(\frac{-2c^2 n}{C_\ell^2}\right) + \exp\left(\frac{-2d^2 n}{C_\ell^2}\right). \tag{24}
$$

*Proof.* It can be observed that

$$p(\mathcal{D}_n) = p(\boldsymbol{x}_1, \boldsymbol{x}_1') \cdots p(\boldsymbol{x}_n, \boldsymbol{x}_n')$$
$$= p(\boldsymbol{x}_1) \cdots p(\boldsymbol{x}_n') p(\boldsymbol{x}_1) \cdots p(\boldsymbol{x}_n').$$

Therefore, the probability measure $\mathbb{P}(\mathfrak{D}_n^-(g))$ can be defined as follows:

$$\mathbb{P}(\mathfrak{D}_n^-(g)) = \int_{\mathcal{D}_n \in \mathfrak{D}_n^-(g)} p(\mathcal{D}_n) \, \mathrm{d}\mathcal{D}_n$$
$$= \int_{\mathcal{D}_n \in \mathfrak{D}_n^-(g)} p(\mathcal{D}_n) \, \mathrm{d}\boldsymbol{x}_1 \cdots \mathrm{d}\boldsymbol{x}_n \, \mathrm{d}\boldsymbol{x}_1' \cdots \mathrm{d}\boldsymbol{x}_n'.$$

When exactly one ConfDiff data pair in $S_n$ is replaced, the change of $\widehat{A}(g), \widehat{B}(g), \widehat{C}(g)$ and $\widehat{D}(g)$ will be no more than $C_\ell/n$. By applying McDiarmid's inequality, we can obtain the following inequalities:

$$\mathbb{P}(\mathbb{E}[\widehat{A}(g)] - \widehat{A}(g) \geq a) \leq \exp\left(\frac{-2a^2 n}{C_\ell^2}\right),$$

$$\mathbb{P}(\mathbb{E}[\widehat{B}(g)] - \widehat{B}(g) \geq b) \leq \exp\left(\frac{-2b^2 n}{C_\ell^2}\right),$$

$$\mathbb{P}(\mathbb{E}[\widehat{C}(g)] - \widehat{C}(g) \geq c) \leq \exp\left(\frac{-2c^2 n}{C_\ell^2}\right),$$

$$\mathbb{P}(\mathbb{E}[\widehat{D}(g)] - \widehat{D}(g) \geq d) \leq \exp\left(\frac{-2d^2 n}{C_\ell^2}\right).$$

Furthermore,

$$\mathbb{P}(\mathfrak{D}_n^-(g)) \leq \mathbb{P}(\widehat{A}(g) \leq 0) + \mathbb{P}(\widehat{B}(g) \leq 0) + \mathbb{P}(\widehat{C}(g) \leq 0) + \mathbb{P}(\widehat{D}(g) \leq 0)$$
$$\leq \mathbb{P}(\widehat{A}(g) \leq \mathbb{E}[\widehat{A}(g)] - a) + \mathbb{P}(\widehat{B}(g) \leq \mathbb{E}[\widehat{B}(g)] - b)$$
$$+ \mathbb{P}(\widehat{C}(g) \leq \mathbb{E}[\widehat{C}(g)] - c) + \mathbb{P}(\widehat{D}(g) \leq \mathbb{E}[\widehat{D}(g)] - d)$$
$$= \mathbb{P}(\mathbb{E}[\widehat{A}(g)] - \widehat{A}(g) \geq a) + \mathbb{P}(\mathbb{E}[\widehat{B}(g)] - \widehat{B}(g) \geq b)$$
$$+ \mathbb{P}(\mathbb{E}[\widehat{C}(g)] - \widehat{C}(g) \geq c) + \mathbb{P}(\mathbb{E}[\widehat{D}(g)] - \widehat{D}(g) \geq d)$$
$$\leq \exp\left(\frac{-2a^2 n}{C_\ell^2}\right) + \exp\left(\frac{-2b^2 n}{C_\ell^2}\right) + \exp\left(\frac{-2c^2 n}{C_\ell^2}\right) + \exp\left(\frac{-2d^2 n}{C_\ell^2}\right),$$

which concludes the proof. $\qquad \square$

Then, the proof of Theorem 5 is given.

*Proof of Theorem 5.* To begin with, we prove the first inequality in Theorem 5.

$$\mathbb{E}[\widetilde{R}_{\mathrm{CD}}(g)] - R(g)$$
$$= \mathbb{E}[\widetilde{R}_{\mathrm{CD}}(g) - \widehat{R}_{\mathrm{CD}}(g)]$$
$$= \int_{\mathcal{D}_n \in \mathfrak{D}_n^+(g)} (\widetilde{R}_{\mathrm{CD}}(g) - \widehat{R}_{\mathrm{CD}}(g)) p(\mathcal{D}_n) \, \mathrm{d}\mathcal{D}_n$$
$$+ \int_{\mathcal{D}_n \in \mathfrak{D}_n^-(g)} (\widetilde{R}_{\mathrm{CD}}(g) - \widehat{R}_{\mathrm{CD}}(g)) p(\mathcal{D}_n) \, \mathrm{d}\mathcal{D}_n$$
$$= \int_{\mathcal{D}_n \in \mathfrak{D}_n^-(g)} (\widetilde{R}_{\mathrm{CD}}(g) - \widehat{R}_{\mathrm{CD}}(g)) p(\mathcal{D}_n) \, \mathrm{d}\mathcal{D}_n \geq 0,$$

where the last inequality is derived because $\widetilde{R}_{\mathrm{CD}}(g)$ is an upper bound of $\widehat{R}_{\mathrm{CD}}(g)$. Furthermore,

$$
\begin{aligned}
&\mathbb{E}[\widetilde{R}_{\mathrm{CD}}(g)] - R(g) \\
&= \int_{\mathcal{D}_n \in \mathfrak{D}_n^-(g)} (\widetilde{R}_{\mathrm{CD}}(g) - \widehat{R}_{\mathrm{CD}}(g)) p(\mathcal{D}_n) \, \mathrm{d}\mathcal{D}_n \\
&\leq \sup_{\mathcal{D}_n \in \mathfrak{D}_n^-(g)} (\widetilde{R}_{\mathrm{CD}}(g) - \widehat{R}_{\mathrm{CD}}(g)) \int_{\mathcal{D}_n \in \mathfrak{D}_n^-(g)} p(\mathcal{D}_n) \, \mathrm{d}\mathcal{D}_n \\
&= \sup_{\mathcal{D}_n \in \mathfrak{D}_n^-(g)} (\widetilde{R}_{\mathrm{CD}}(g) - \widehat{R}_{\mathrm{CD}}(g)) \mathbb{P}(\mathfrak{D}_n^-(g)) \\
&= \sup_{\mathcal{D}_n \in \mathfrak{D}_n^-(g)} (f(\widehat{A}(g)) + f(\widehat{B}(g)) + f(\widehat{C}(g)) + f(\widehat{D}(g)) \\
&\quad - \widehat{A}(g) - \widehat{B}(g) - \widehat{C}(g) - \widehat{D}(g)) \mathbb{P}(\mathfrak{D}_n^-(g)) \\
&\leq \sup_{\mathcal{D}_n \in \mathfrak{D}_n^-(g)} (L_f|\widehat{A}(g)| + L_f|\widehat{B}(g)| + L_f|\widehat{C}(g)| + L_f|\widehat{D}(g)| \\
&\quad + |\widehat{A}(g)| + |\widehat{B}(g)| + |\widehat{C}(g)| + |\widehat{D}(g)|) \mathbb{P}(\mathfrak{D}_n^-(g) \\
&= \sup_{\mathcal{D}_n \in \mathfrak{D}_n^-(g)} \frac{L_f+1}{2n}(|\sum_{i=1}^n (\pi_+ - c_i)\ell(g(\boldsymbol{x}_i),+1)| + |\sum_{i=1}^n (\pi_- - c_i)\ell(g(\boldsymbol{x}_i'),-1)| \\
&\quad + |\sum_{i=1}^n (\pi_+ + c_i)\ell(g(\boldsymbol{x}_i'),+1)| + |\sum_{i=1}^n (\pi_- + c_i)\ell(g(\boldsymbol{x}_i),-1)|) \mathbb{P}(\mathfrak{D}_n^-(g)) \\
&\leq \sup_{\mathcal{D}_n \in \mathfrak{D}_n^-(g)} \frac{L_f+1}{2n}(\sum_{i=1}^n |(\pi_+ - c_i)\ell(g(\boldsymbol{x}_i),+1)| + \sum_{i=1}^n |(\pi_- - c_i)\ell(g(\boldsymbol{x}_i'),-1)| \\
&\quad + \sum_{i=1}^n |(\pi_+ + c_i)\ell(g(\boldsymbol{x}_i'),+1)| + \sum_{i=1}^n |(\pi_- + c_i)\ell(g(\boldsymbol{x}_i),-1)|) \mathbb{P}(\mathfrak{D}_n^-(g)) \\
&= \sup_{\mathcal{D}_n \in \mathfrak{D}_n^-(g)} \frac{L_f+1}{2n} \sum_{i=1}^n (|(\pi_+ - c_i)\ell(g(\boldsymbol{x}_i),+1)| + |(\pi_- - c_i)\ell(g(\boldsymbol{x}_i'),-1)| \\
&\quad + |(\pi_+ + c_i)\ell(g(\boldsymbol{x}_i'),+1)| + |(\pi_- + c_i)\ell(g(\boldsymbol{x}_i),-1)|) \mathbb{P}(\mathfrak{D}_n^-(g)) \\
&\leq \sup_{\mathcal{D}_n \in \mathfrak{D}_n^-(g)} \frac{(L_f+1)C_\ell}{2n} \sum_{i=1}^n (|\pi_+ - c_i| + |\pi_- - c_i| + |\pi_+ + c_i| + |\pi_- + c_i|) \mathbb{P}(\mathfrak{D}_n^-(g)).
\end{aligned}
$$

Similar to the proof of Theorem 3, we can obtain

$$
|\pi_+ - c_i| + |\pi_- - c_i| + |\pi_+ + c_i| + |\pi_- + c_i| \leq 4.
$$

Therefore, we have

$$
\mathbb{E}[\widetilde{R}_{\mathrm{CD}}(g)] - R(g) \leq 2(L_f+1)C_\ell \Delta,
$$

which concludes the proof of the first inequality in Theorem 5. Before giving the proof of the second inequality, we give the upper bound of $|\widetilde{R}_{\mathrm{CD}}(g) - \mathbb{E}[\widetilde{R}_{\mathrm{CD}}(g)]|$. When exactly one ConfDiff data pair in $\mathcal{D}_n$ is replaced, the change of $\widetilde{R}_{\mathrm{CD}}(g)$ is no more than $2C_\ell L_f/n$. By applying McDiarmid's inequality, we have the following inequalities with probability at least $1 - \delta/2$:

$$
\widetilde{R}_{\mathrm{CD}}(g) - \mathbb{E}[\widetilde{R}_{\mathrm{CD}}(g)] \leq 2C_\ell L_f \sqrt{\frac{\ln 2/\delta}{2n}},
$$

$$
\mathbb{E}[\widetilde{R}_{\mathrm{CD}}(g)] - \widetilde{R}_{\mathrm{CD}}(g) \leq 2C_\ell L_f \sqrt{\frac{\ln 2/\delta}{2n}}.
$$

Therefore, with probability at least $1 - \delta$, we have

$$
|\widetilde{R}_{\mathrm{CD}}(g) - \mathbb{E}[\widetilde{R}_{\mathrm{CD}}(g)]| \leq 2C_\ell L_f \sqrt{\frac{\ln 2/\delta}{2n}}.
$$

Finally, we have

$$
\begin{aligned}
|\widetilde{R}_{\mathrm{CD}}(g) - R(g)| &= |\widetilde{R}_{\mathrm{CD}}(g) - \mathbb{E}[\widetilde{R}_{\mathrm{CD}}(g)] + \mathbb{E}[\widetilde{R}_{\mathrm{CD}}(g)] - R(g)| \\
&\leq |\widetilde{R}_{\mathrm{CD}}(g) - \mathbb{E}[\widetilde{R}_{\mathrm{CD}}(g)]| + |\mathbb{E}[\widetilde{R}_{\mathrm{CD}}(g)] - R(g)| \\
&= |\widetilde{R}_{\mathrm{CD}}(g) - \mathbb{E}[\widetilde{R}_{\mathrm{CD}}(g)]| + \mathbb{E}[\widetilde{R}_{\mathrm{CD}}(g)] - R(g) \\
&\leq 2C_\ell L_f \sqrt{\frac{\ln 2/\delta}{2n}} + 2(L_f + 1)C_\ell \Delta,
\end{aligned}
$$

with probability at least $1 - \delta$, which concludes the proof. $\qquad\square$

## F  Proof of Theorem 6

With probability at least $1 - \delta$, we have

$$
\begin{aligned}
R(\widetilde{g}_{\mathrm{CD}}) - R(g^*) =& (R(\widetilde{g}_{\mathrm{CD}}) - \widetilde{R}_{\mathrm{CD}}(\widetilde{g}_{\mathrm{CD}})) + (\widetilde{R}_{\mathrm{CD}}(\widetilde{g}_{\mathrm{CD}}) - \widetilde{R}_{\mathrm{CD}}(\widehat{g}_{\mathrm{CD}})) \\
& + (\widetilde{R}_{\mathrm{CD}}(\widehat{g}_{\mathrm{CD}}) - R(\widehat{g}_{\mathrm{CD}})) + (R(\widehat{g}_{\mathrm{CD}}) - R(g^*)) \\
\leq& |R(\widetilde{g}_{\mathrm{CD}}) - \widetilde{R}_{\mathrm{CD}}(\widetilde{g}_{\mathrm{CD}})| + |\widetilde{R}_{\mathrm{CD}}(\widehat{g}_{\mathrm{CD}}) - R(\widehat{g}_{\mathrm{CD}})| + (R(\widehat{g}_{\mathrm{CD}}) - R(g^*)) \\
\leq& 4C_\ell(L_f + 1)\sqrt{\frac{\ln 2/\delta}{2n}} + 4(L_f + 1)C_\ell\Delta + 8L_\ell \mathfrak{R}_n(\mathcal{G}).
\end{aligned}
$$

The first inequality is derived because $\widetilde{g}_{\mathrm{CD}}$ is the minimizer of $\widetilde{R}_{\mathrm{CD}}(g)$. The second inequality is derived from Theorem 5 and Theorem 3. The proof is completed. $\qquad\square$

## G  Limitations and Potential Negative Social Impacts

### G.1  Limitations

This work focuses on binary classification problems. To generalize it to multi-class problems, we need to convert multi-class classification to a set of binary classification problems via the one-versus-rest or the one-versus-one strategies. In the future, developing methods directly handling multi-class classification problems is promising.

### G.2  Potential Negative Social Impacts

This work is within the scope of weakly supervised learning, which aims to achieve comparable performance while reducing labeling costs. Therefore, when this technique is very effective and prevalent in society, the demand for data annotations may be reduced, leading to the increasing unemployment rate of data annotation workers.

## H  Additional Information about Experiments

In this section, the details of experimental data sets and hyperparameters are provided.

### H.1  Details of Experimental Data Sets

The detailed statistics and corresponding model architectures are summarized in Table 4. The basic information of data sets, sources and data split details are elaborated as follows.

For the four benchmark data sets,

- MNIST [61]: It is a grayscale handwritten digits recognition data set. It is composed of 60,000 training examples and 10,000 test examples. The original feature dimension is 28*28, and the label space is 0-9. The even digits are regarded as the positive class while the odd digits are regarded as the negative class. We sampled 15,000 unlabeled data pairs as training data. The data set can be downloaded from http://yann.lecun.com/exdb/mnist/.

Table 4: Characteristics of experimental data sets.

| Data Set | # Train | # Test | # Features | # Class Labels | Model |
|---|---|---|---|---|---|
| **MNIST** | 60,000 | 10,000 | 784 | 10 | MLP |
| **Kuzushiji** | 60,000 | 10,000 | 784 | 10 | MLP |
| **Fashion** | 60,000 | 10,000 | 784 | 10 | MLP |
| **CIFAR-10** | 50,000 | 10,000 | 3,072 | 10 | ResNet-34 |
| **Optdigits** | 4,495 | 1,125 | 62 | 10 | MLP |
| **USPS** | 7,437 | 1,861 | 256 | 10 | MLP |
| **Pendigits** | 8,793 | 2,199 | 16 | 10 | MLP |
| **Letter** | 16,000 | 4,000 | 16 | 26 | MLP |

- Kuzushiji-MNIST [62]: It is a grayscale Japanese character recognition data set. It is composed of 60,000 training examples and 10,000 test examples. The original feature dimension is 28*28, and the label space is {'o', 'su','na', 'ma', 're', 'ki','tsu','ha', 'ya','wo'}. The positive class is composed of 'o', 'su','na', 'ma', and 're' while the negative class is composed of 'ki','tsu','ha', 'ya', and 'wo'. We sampled 15,000 unlabeled data pairs as training data. The data set can be downloaded from `https://github.com/rois-codh/kmnist`.
- Fashion-MNIST [63]: It is a grayscale fashion item recognition data set. It is composed of 60,000 training examples and 10,000 test examples. The original feature dimension is 28*28, and the label space is {'T-shirt', 'trouser', 'pullover', 'dress', 'sandal', 'coat', 'shirt', 'sneaker', 'bag', 'ankle boot'}. The positive class is composed of 'T-shirt', 'pullover', 'coat', 'shirt', and 'bag' while the negative class is composed of 'trouser', 'dress', 'sandal', 'sneaker', and 'ankle boot'. We sampled 15,000 unlabeled data pairs as training data. The data set can be downloaded from `https://github.com/zalandoresearch/fashion-mnist`.
- CIFAR-10 [64]: It is a colorful object recognition data set. It is composed of 50,000 training examples and 10,000 test examples. The original feature dimension is 32*32*3, and the label space is {'airplane', 'bird', 'automobile', 'cat', 'deer', 'dog', 'frog', 'horse', 'ship', 'truck'}. The positive class is composed of 'bird', 'deer', 'dog', 'frog', 'cat', and 'horse' while the negative class is composed of 'airplane', 'automobile', 'ship', and 'truck'. We sampled 10,000 unlabeled data pairs as training data. The data set can be downloaded from `https://www.cs.toronto.edu/~kriz/cifar.html`.

For the four UCI data sets, they can be downloaded from Dua and Graff [65].

- Optdigits, USPS, Pendigits [65]: They are handwritten digit recognition data set. The train-test split can be found in Table 4. The feature dimensions are 62, 256, and 16 respectively and the label space is 0-9. The even digits are regarded as the positive class while the odd digits are regarded as the negative class. We sampled 1,200, 2,000, and 2,500 unlabeled data pairs for training respectively.
- Letter [65]: It is a letter recognition data set. It is composed of 16,000 training examples and 4,000 test examples. The feature dimension is 16 and the label space is the 26 capital letters in the English alphabet. The positive class is composed of the top 13 letters while the negative class is composed of the latter 13 letters. We sampled 4,000 unlabeled data pairs for training.

## H.2 Details of Experiments on the KuaiRec Data Set

We used the small matrix of the KuaiRec [72] data set since it has dense confidence scores. It has 1,411 users and 3,327 items. We clipped the watching ratio above 2 and regarded the examples with watching ratio greater than 2 as positive examples. Following the experimental protocol of He et al. [73], we regarded the latest positive example foe each user as the positive testing data, and sampled 49 negative testing data to form the testing set for each user. The HR and NDCG were calculated at top 10. The learning rate was set to 1e-3 and the dropout rate was set to 0.5. The number of epochs was set to 50 and the batch size was set to 256. The number of MLP layers was 2 and the embedding dimension was 128. The hyperparameters was the same for all the approaches for a fair comparison.

### H.3 Details of Hyperparameters

For MNIST, Kuzushiji-MNIST and Fashion-MNIST, the learning rate was set to 1e-3 and the weight decay was set to 1e-5. The batch size was set to 256 data pairs. For training the probabilistic classifier to generate confidence, the batch size was set to 256 and the epoch number was set to 10.

For CIFAR10, the learning rate was set to 5e-4 and the weight decay was set to 1e-5. The batch size was set to 128 data pairs. For training the probabilistic classifier to generate confidence, the batch size was set to 128 and the epoch number was set to 10.

For all the UCI data sets, the learning rate was set to 1e-3 and the weight decay was set to 1e-5. The batch size was set to 128 data pairs. For training the probabilistic classifier to generate confidence, the batch size was set to 128 and the epoch number was set to 10.

The learning rate and weight decay for training the probabilistic classifier were the same as the setting for each data set correspondingly.

## I   More Experimental Results with Fewer Training Data

Figure 4 shows extra experimental results with fewer training data on other data sets with different class priors.

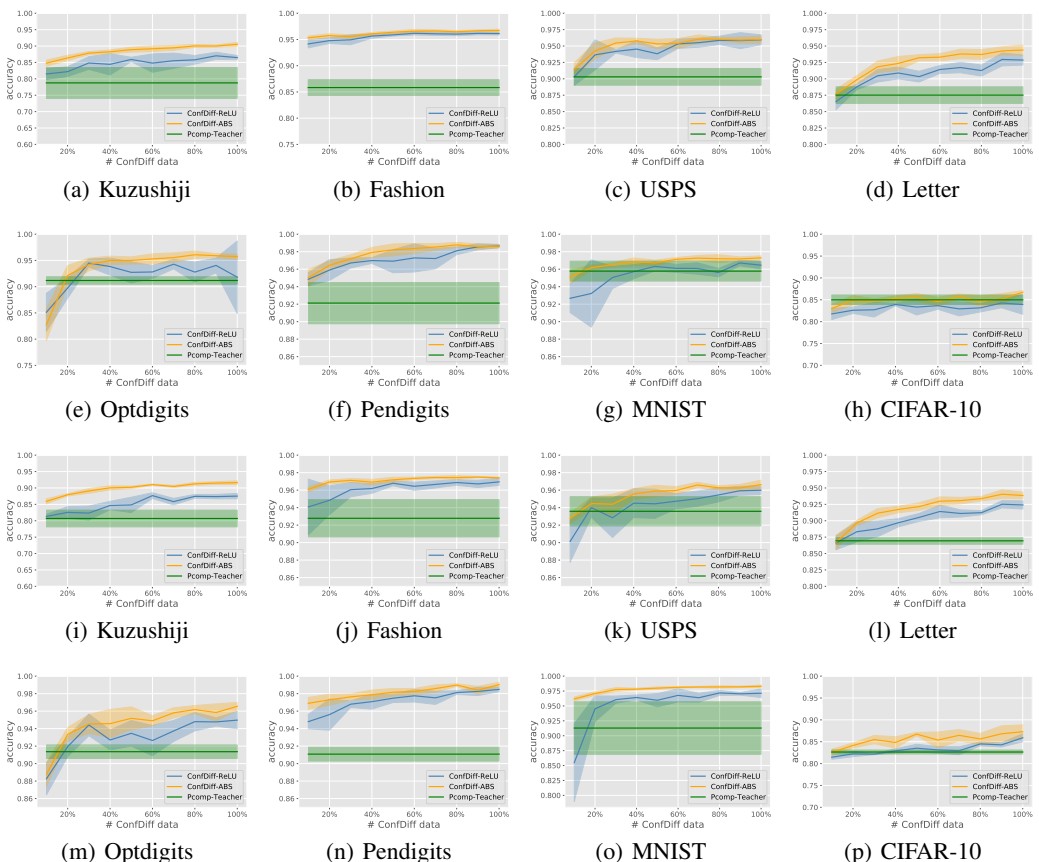

Figure 4: Classification performance of ConfDiff-ReLU and ConfDiff-ABS given a fraction of training data as well as Pcomp-Teacher given 100% of training data with different prior settings ($\pi_+ = 0.2$ for the fist and the second row and $\pi_+ = 0.8$ for the third and the fourth row).

