# OpenReview forum: "Binary Classification with Confidence Difference"
_NeurIPS.cc/2023/Conference — NeurIPS 2023 poster_

### Official Review · Reviewer_G7aZ · 2023-07-06

**Soundness:** 4 excellent
**Presentation:** 4 excellent
**Contribution:** 4 excellent
**Rating:** 9
**Confidence:** 3

**Summary:**

This paper studies binary classification problems. A new data type is introduced, where each observation consists of two input instances, together with the "confidence difference," defined as the difference of the conditional probabilities (output=1|input) of the two input instances. New loss functions are introduced and the excess risk bound is derived with probability. To extend the research, the author(s) also studied the case when the confidence difference is contaminated by noise, and the idea of rectifying the empirical loss function by dropping the negative summands. Empirical tests are done on benchmark datasets where the proposed algorithm consistently outperforms the competitors.

**Strengths:**

To my knowledge, the proposed data type (based on confidence difference) is new. The presented research is original. The high quality of the research is guaranteed by solid theoretical analysis, and the definition of a new class of classification data that captures more possible application scenarios. This paper is written clearly and is easy to follow. The broad application scenarios support the significance of the research.

**Weaknesses:**

I have no major concerns about this work. Some minor issues are listed in the "Questions" section below, for the author(s) to clarify.

**Questions:**

1. Table 2 has a structure similar to Table 1 in [11]. In particular, the methods Pcomp-*** have different values from those reported in Table 1 in [11]. Would it be possible for the author(s) to take a look at Table 1 in [11] and briefly explain the difference in experiment design or other conditions that leads to such a difference in the values in these two tables? Same question for Table 2 of this manuscript, and Table 3 in [11].
2. It seems that the density (or probability mass function?) tilde-p defined in Eq. (4) is not used elsewhere. Is that true? Is there any insight into this density?
3. Since x and x' are drawn from the same probability distribution, what is the relation between x_i and x_i'? Is the sample in Line 137 just n independent copies? Would the analysis still work if n=m^2 is the square of some integer, so that x_i and x_i's are obtained by the full combination of some sample points u_1,...,u_m?
4. Is the definition (6) and (8) newly introduced, or cited from the literature? It would be better if the author(s) could specify the facts around these equations.

**Limitations:**

This is theoretical research, and I have not identified any limitations.

---

> ### Author Rebuttal · Authors · 2023-08-09
>
> Thank you for reviewing our paper. We are encouraged that you agree with the novelty and contributions of our paper. Below are the answers to your questions.
>
> ***
> **Q1: Difference in experimental design between this paper and [11].**
>
> **A1:** In [11], they only used examples from ${(+1, +1),(+1, -1),(-1, -1)}$, while ours also used examples from $(-1, +1)$. Because the training data is different, the experimental results are different. In order to make a fair comparison, we adapted the Pcomp methods to our problem settings. In particular, the input to Pcomp methods can be data pairs where the first example is more likely to be positive. Therefore, when the confidence difference was greater than zero, we used this data pair directly. If the confidence difference was less than zero, we swapped them. We will include the details of the experimental settings in the final version of our paper.
>
> ***
> **Q2: The density in Eq.(4) is not used and what is the insight?**
>
> **A2:** Yes, we only used Eq.(4) to elaborate the data distribution assumption of the Pcomp methods. As discussed in the Introduction section, Pcomp methods only consider examples from ${(+1, +1),(+1, -1),(-1, -1)}$. To handle training data with labels $(-1, +1)$, they have to discard them or reverse them to $(+1, -1)$. On the contrary, our data distribution assumption is more general, because we explicitly consider the examples from $(-1, +1)$ in the data distribution assumption.
>
> ***
> **Q3: The relationship between $(x, x')$ and whether the analysis will still work for data pairs of size $m^2$ obtained from combining unlabeled data of size $m$.**
>
> **A3:** First, we assume that $x$ and $x'$ in an unlabeled data pair are i.i.d. and both sampled from $p(x)$. Therefore, based on our assumption, any two unlabeled data pairs are also *mutually independent*. If we use all pairwise combinations of an unlabeled data set, some of the resulting data pairs may not be i.i.d. For example, we may have two unlabeled data pairs $(x_1, x_2)$ and $(x_1, x_3)$ both in the training set. Because they contain the same example $x_1$, I am afraid that they are *dependent*. Therefore, our theoretical analysis may not apply to this setting, and we need to investigate new theories that formulate non-i.i.d. training data. Thank you for the insightful question and we will include the discussion in our paper.
>
> ***
> **Q4: Are Equations (6) and (8) newly introduced and what are the facts about them?**
>
> **A4:** The two equations are new. Eq.(6) is the definition of confidence difference and Eq.(8) is the empirical approximation of Eq.(7), also known to be an *unbiased risk estimator*. We introduced Eq.(6) to specify the confidence difference between two unlabeled examples, which can also measure how confident the pairwise comparison is. Eq.(8) is a natural result of using training data to estimate the expected value. In Theorem 3, we discussed the estimation error bound by using the unbiased risk estimator. We will add this discussion to our paper.

---

> > ### Comment · Reviewer_G7aZ · 2023-08-16
> > **notation not used elsewhere may be removed**
> >
> > We appreciate the comprehensive information.
> >
> > For Q2, we think that a notation not used elsewhere may be removed.

---

> > > ### Author Response · Authors · 2023-08-16
> > >
> > > Thank you for your suggestion. We will remove the unused notation in the final version of our paper.

---

### Official Review · Reviewer_XYQ6 · 2023-07-06

**Soundness:** 3 good
**Presentation:** 4 excellent
**Contribution:** 2 fair
**Rating:** 6
**Confidence:** 3

**Summary:**

The paper discusses weak supervised learning for binary classification, specially in the setting where labeled data is not available. Based on pairwise-comparison (Pcomp) confidence, where we are given data pair $(x_1, x_2)$ and binary label {+1, -1} of $x_1$ being more or less probable of being positive compared to $x_2$, this paper suggests the novel problem setting of ConfDiff, where we are given $((x_1, x_2), P(y = 1 | x_2) - P(y = 1 | x_1))$. The paper then proceeds to provide risk estimators and error bounds for this new problem setting. Experimental results on curated benchmarks and one real benchmark shows the usefulness of the paper’s theoretical results and motivated algorithm.

**Strengths:**

1. The paper is well written and organized.
2. The theoretical results in the paper are non-trivial and well done.
3. Good experiments and ablation studies.


**Weaknesses:**

My main complain with the paper is that the problem setting does not feel well justified. Since the main goal of the paper is to suggest a novel problem setting, this is a big weakness.

For example, in line 39, the paper makes the claim that $P(y = 1 | x_2) - P(y = 1 | x_1)$ values might be less biased than individual $P(y = 1 | x_2)$ and  $P(y = 1 | x_1)$ estimates by human labelers. I do not think this claim is well justified.

Similarly, line 51 claims that section 4 will give a real world case study of this problem. While section 4 does present a real world problem where the paper’s algorithm works better, it is not clear that this is due to the problem setup being more useful/ideal. More study would need to establish that this problem setup has indeed lower bias then the case of pointwise confidence value, possibly with collecting a large scale dataset with human annotators.

Finally, if someone is unconvinced of the problem setting, the paper seems **tautological**, i.e., defining a custom problem where a modified risk estimator works (which, while the proof being involved, is not hard to see).


**Questions:**

1. Based on the weakness mentioned, are there prior works that suggests this problem setup is more meaningful?
2. Line 233, “Since the data sets were originally designed for multi-class classification, we manually partitioned them into binary classes.”, how is this manual partitioning done for each dataset? Referencing to the appendix would be fine here.

---

> ### Author Rebuttal · Authors · 2023-08-09
>
> First, we are very grateful for your time and effort in reviewing this paper. Below are the responses to your questions and comments.
>
> ***
> **Q1: The claim that the confidence difference has a lower bias is not well justified.**
>
> **A1:** Thank you for your comment. As discussed in the Introduction section, the confidence difference can be less biased in *some applications*. Take the problem of disease risk estimation as an example. Different doctors with different backgrounds and experience will give different confidence scores given a person's attributes. Some are confident and tend to assign high confidence, while others are relatively conservative and tend to assign low confidence. As a result, considering them all together in the training set may have negative effects. In such cases, using the confidence difference may be a more objective statistic by describing the relative difference between two examples for a doctor. We agree with you that using more large real-world data sets will better motivate. We will consider collecting more large-scale data sets using crowdsourcing platforms (e.g. MTurk) as our future work.
>
> ***
> **Q2: Are the advantages in the experimental results of the recommender system task the result of better settings? Is there any prior work suggesting that the setting is more meaningful?**
>
> **A2:** In this experiment, our approach performs better in terms of HR and achieves comparable performance in terms of NDCG compared with the Pconf method. Since our approach uses less supervision information, the Pconf method should have performed better than ours. Therefore, we suspect that this is because Pconf is affected by biased pointwise confidence values, while ConfDiff can deal with this problem in a better way.
>
> As positive-confidence classification is a relatively new problem, there is no previous work on this setting. Therefore, the setting is new to the literature. We agree with you that more case studies will further support our setting. We will dedicate ourselves to collecting more large-scale real-world data sets to further validate the superiority of our setting.
> ***
> **Q3: Partition of data sets.**
>
> **A3:** We have listed the details of the data set partition in Appendix I. We will add the reference to our paper.

---

> > ### Comment · Reviewer_XYQ6 · 2023-08-11
> >
> > Follow-up question: **The claim that the confidence difference has a lower bias is not well justified.**
> >
> > Based on the authors' answer, do all the data points $((x, x'), c(x, x'))$ get collected from the same expert/confidence estimation system? The medical use-case example the authors provided works if the confidence estimate of two doctors, let them be $c_1(x)$ and $c_2(x)$, vary by a constant, i.e, there exists $A$ such that $c_1(x) = c_2(x) + A$ for all $x$. Then for any $x, x'$, we have $c_1(x) - c_1(x') = c_2(x) - c_2(x')$.
> >
> > However, it is quite possible that this is not the case. In the extreme case, the confidence difference estimated by two doctors on the very same data pair can be different. In the less extreme case, similar data point pair can get different confidence difference estimate by two doctors. It is quite likely that any such large scale dataset would contain annotations by multiple experts, thereby bringing the same bias.
> >
> > Would the authors have any intuitive or theoretical explanation for why confidence difference would have a lower bias in this case?

---

> > > ### Author Response · Authors · 2023-08-12
> > > **Response to the Further Question**
> > >
> > > First of all, we would like to express our sincere gratitude for your valuable comments and insights. Based on your valuable question, both pointwise confidence and pairwise confidence difference can be biased in many cases. Then, we conducted more theoretical analysis and found that our setting would have a lower bias.
> > >
> > > We assume that the output of the confidence estimation system is given by a deterministic function. Let $x$ denote the ground-truth pointwise confidence value and $y$ denote the output confidence value given by the estimation system.
> > >
> > > For a pair of examples, the ground-truth confidence values are $(x_1, x_2)$. The outputs of the confidence estimation system are $(y_1, y_2)$. If we use the pointwise confidence value directly, the total bias is $$E_{point}=|y_1-x_1|+|y_2-x_2|.$$
> > > If we consider the confidence difference, then the total bias is
> > > $$
> > > \begin{align}
> > > E_{pair}&=|(y_1-y_2)-(x_1-x_2)| \\\\
> > > &=|(y_1-x_1)-(y_2-x_2)|.
> > > \end{align}
> > > $$
> > > Based on the absolute value inequality that $|A-B|\leq |A|+|B|$, we have $E_{pair} \leq E_{point}$. The implication is that the pairwise confidence difference can have a lower bias than the pointwise confidence for a given estimation system. The conclusion can also be extended to the existence of multiple estimation systems (doctors). Therefore, the confidence difference would have a lower bias in this case.
> > >
> > > I hope this rebuttal addresses your concerns. If you have any further concerns or questions, please do not hesitate to raise them. Thank you again for your time and effort in reviewing our submission.

---

### Official Review · Reviewer_DX8U · 2023-07-06

**Soundness:** 3 good
**Presentation:** 3 good
**Contribution:** 3 good
**Rating:** 6
**Confidence:** 3

**Summary:**

- This work proposes a confidence label based training approach called _ConfDiff_ for binary classification models as an improvement over traditional hard labels. Specifically, authors argue that obtaining hard labels for traditional supervised learning paradigm, or confidence metrics around positive labels for weakly supervised learning paradigm is expensive and at times infeasible process. Therefore, they propose training such classification models on pairwise comparisons between confidence in positive labeling of training data example.

- Authors prove error bounds, and demonstrate the robustness of the proposed approach theoretically.

- Authors evaluate the proposed approach on benchmark classification datasets and a real world recommender system.



**Strengths:**

**Motivation**
Authors make a strong technical as well as intuitive case in support of their approach, _ConfDiff_, in comparison to prevalent approaches in the related work such as _Pconf_. Similar to traditional supervised learning paradigm where approaches such as _soft labeling_, _pairwise loss_, etc. have provided benefits over pointwise models utilizing hard labels in the form of increased robustness and robustness, it is intuitive that pairwise confidence differences should outperform methods like _Pconf_, as demonstrated by the paper.

**Technical Presentation**
The work is very well presented and easy to digest. The Preliminaries section sets up the reader very well equipped to compare/contrast the proposed approach with existing classification approaches. Specifically, the authors do a great job at using formal notations with perfect granularity required to compare different approaches, without being overwhelming to the reader.

**Experiments**
Authors evaluate their proposed approach on multiple benchmark datasets as well one real world recommender system dataset. All the details as well as source code required for reproducibility are available in the Appendix section of the work. The observed evaluation metrics clearly favors the proposed approach against the baseline comparisons. Authors further analyze the sample efficiency and robustness of the proposed approach and provide empirical evidence supporting the theoretical analysis in the prior sections.

**Weaknesses:**

**Choice of comparative baselines**
While the authors clearly position their work as an improvement over the existing approaches in weakly supervised learning domain, they make a case throughout the introductory as well as technical sections regarding benefits over traditional hard label supervised learning. Given that _ConfDiff_ was evaluated using backbones such as ResNet-34, I believe it would help the work to include inside Table 1 the metrics corresponding to vanilla ResNet-34 or logistic regressions.

**Generalizability of proposed approach**
Theoretical contributions not withstanding, it is not entirely clear how generalizable the proposed approach is in terms of obtaining confidence difference values. Authors present the real world evaluation on  KauiRec dataset where _watching ratio_ i.e., ratio of watching time of short video and total length of short video, has an intuitive relationship with $p(y=1|x)$.
> We clipped the watching ratio above 2 and regarded the examples with watching ratio greater than 2 as positive examples. Following the experimental protocol of [8], we regarded the latest positive example for each user as the positive testing data, and sampled 49 negative testing data to form the testing set for each user.

As I understand, based on the above text in the Appendix, even in this example, the authors had to resort to assigning hard labels to training data using a heuristic based approach. I am not sure what would be a similar intuitive approach to compute $c(x, x^{\prime})$ in a more common user-item recommender system where purchase labels/click labels are by nature hard labels, and the priors are heavily biased by the fact that only limited number of recommendations can be shown to a user?

**Questions:**

- Could you address the weakness/include discussions about the same in the manuscript?

**Limitations:**

In the Appendix section, authors have addressed limitations of the approach being currently limited to binary classification and potential negative societal impact in terms of loss of annotation work.

---

> ### Author Rebuttal · Authors · 2023-08-09
>
> First of all, we are very grateful for your time and effort in reviewing this submission. We are encouraged that you agree with the contributions of our paper. Below are the responses to your comments.
> ***
> **Q1: Add experimental results from the vanilla ResNet.**
>
> **A1:** We agree with you that adding experimental results of logistic regression (LR) will strengthen the paper. We will add the following experimental results to Table 1 and Table 2.
>
> | Class Prior   | Method     | MNIST       | Kuzushiji   | Fashion     | CIFAR-10    |
> | ------------- | ---------- | ----------- | ----------- | ----------- | ----------- |
> | $\pi_{+}=0.2$ | Supervised | 0.990±0.000 | 0.939±0.001 | 0.979±0.001 | 0.894±0.003 |
> | $\pi_{+}=0.5$ | Supervised | 0.986±0.000 | 0.929±0.002 | 0.976±0.001 | 0.871±0.003 |
> | $\pi_{+}=0.8$ | Supervised | 0.991±0.001 | 0.942±0.003 | 0.979±0.000 | 0.897±0.002 |
>
> | Class Prior   | Method     | Optdigits   | USPS        | Pendigits   | Letter      |
> | ------------- | ---------- | ----------- | ----------- | ----------- | ----------- |
> | $\pi_{+}=0.2$ | Supervised | 0.990±0.002 | 0.984±0.002 | 0.997±0.001 | 0.978±0.003 |
> | $\pi_{+}=0.5$ | Supervised | 0.988±0.003 | 0.980±0.003 | 0.997±0.001 | 0.975±0.001 |
> | $\pi_{+}=0.8$ | Supervised | 0.987±0.003 | 0.983±0.002 | 0.997±0.001 | 0.976±0.004 |
>
> The performance of LR is comparable to the Pconf method, which can serve as a reference for performance comparisons.
> ***
> **Q2: The application to common recommender system tasks even with biased class priors.**
>
> **A2:** Currently, our approaches require supervision information of *confidence labels*. Therefore, if there are only 0-1 interactions without any other side information, it may be difficult to directly apply our methods without *confidence labels*. We will include this limitation of our applications in our paper. However, we can still obtain such confidence labels by using an auxiliary probabilistic classifier that outputs the probability of being positive, which is often used for many recommender system problems, such as click-through rate (CTR) prediction (Zhou et al., 2018). In addition, for many real-world recommender system applications, such as news recommendation, short video recommendation, and movie recommendation, we can often collect some real-value *confidence labels*, such as watching ratios and user ratings. We have also discussed the influence of inaccurate confidence values on our approach in Section 3.3. Therefore, our method is promising to be applied to more recommender system problems.
>
> We agree with you that the class prior may be biased from the training data to the testing data. Such a problem is the well-known *label shift* problem in the distribution shift literature (Lipton et al., 2018). In this paper, we only consider that the ordinally labelled training data and testing data are sampled from the same distribution. We will consider the development of approaches that address label shift as our future work. Furthermore, if the class prior is the same for training and testing and the estimated class prior is biased, we have discussed its influence with theoretical analysis in Section 3.3. It is shown that a more accurate estimation of the class prior will reduce the upper bound of the estimation error and facilitate model training.
>
> ***
>
> Reference:
>
> - Zhou et al., Deep interest network for click-through rate prediction, KDD 2018.
> - Lipton et al., Detecting and correcting for label shift with black box predictors, ICML 2018.

---

> > ### Comment · Reviewer_DX8U · 2023-08-14
> >
> > I thank the authors for addressing some of my concerns mentioned in the review and their acknowledgement of a potential limitation of the proposed approach.
> >
> > Considering the additional information, my overall rating of the work remains unchanged.

---

### Official Review · Reviewer_Qs9H · 2023-07-07

**Soundness:** 3 good
**Presentation:** 3 good
**Contribution:** 2 fair
**Rating:** 6
**Confidence:** 4

**Summary:**

The paper proposes to solve a classification problem using weakly supervised learning problem called confidence-difference (ConfDiff)
classification, where unlabeled data pairs are equipped with confidence difference specifying the difference in the probabilities of being positive. The authors further develop a risk-consistent approach to tackle this problem and show that the estimation error bound achieves the optimal convergence rate. They provide additional analysis for noisy labels.  The authors empirically show the effectiveness of their suggested technique on several classification benchmarks such as MNIST, CIFAR-10 and FASHION datasets (TABLE 1), where they show that they approach performs better compared to PComp methods. They further show effectiveness of their method  in leveraging the supervision information of the confidence difference on a real-world recommender system data set.

**Strengths:**

The paper is relatively clear and presents an interesting take on classification approach. Instead of an exact label the paper proposes to use relative confidence and provides theory to establish the performance bounds. The paper further validates the effectiveness  on a real world recommender dataset, surpassing PComp teacher. The authors also investigate what happens when the volume of the labeled data is reduced and show that the suggested approach achieves superior or comparable performance even when only 10% of training data are used. It elucidates that leveraging confidence difference may be more effective than increasing the number of training examples.

**Weaknesses:**

My main concern is that there is no comparison to a performance based on regular labels. Such comparison can at least show the gap in case regular classification performs better (and labels can be provided). Also the authors mention medical application in motivation, however they do not provide experiments with similar data.

**Questions:**

Please add performance of regular classification for your experiments if such is possible to benchmark.
Are there any other methods applicable for the Recommendation systems you can compare against? Providing more comparisons will help to strengthen your experimental section.

**Limitations:**

I feel there are not so many applications that will have relative comparison between two samples. The two applications provided in the motivation are well-suited for such approach, but unfortunately the method is only benchmarked on one application. The authors might want to benchmark against additional application that benefits from relative comparison.

---

> ### Author Rebuttal · Authors · 2023-08-09
>
> First, we would like to thank you for your time and effort in reviewing our submission. Next, we would like to respond to the main concerns raised in the comments.
> ***
> **Q1: Comparison with supervised learning based on ordinary labels.**
>
> **A1:** We agree with you and list the performance of the supervised learning method.
> Here are the experimental results on the four benchmark data sets:
>
> | Class Prior   | Method     | MNIST       | Kuzushiji   | Fashion     | CIFAR-10    |
> | ------------- | ---------- | ----------- | ----------- | ----------- | ----------- |
> | $\pi_{+}=0.2$ | Supervised | 0.990±0.000 | 0.939±0.001 | 0.979±0.001 | 0.894±0.003 |
> | $\pi_{+}=0.5$ | Supervised | 0.986±0.000 | 0.929±0.002 | 0.976±0.001 | 0.871±0.003 |
> | $\pi_{+}=0.8$ | Supervised | 0.991±0.001 | 0.942±0.003 | 0.979±0.000 | 0.897±0.002 |
>
> Here are the experimental results on the four UCI data sets:
>
> | Class Prior   | Method     | Optdigits   | USPS        | Pendigits   | Letter      |
> | ------------- | ---------- | ----------- | ----------- | ----------- | ----------- |
> | $\pi_{+}=0.2$ | Supervised | 0.990±0.002 | 0.984±0.002 | 0.997±0.001 | 0.978±0.003 |
> | $\pi_{+}=0.5$ | Supervised | 0.988±0.003 | 0.980±0.003 | 0.997±0.001 | 0.975±0.001 |
> | $\pi_{+}=0.8$ | Supervised | 0.987±0.003 | 0.983±0.002 | 0.997±0.001 | 0.976±0.004 |
>
> In particular, the performance is comparable to the Pconf method, which can be used as a reference for performance comparisons.
> ***
> **Q2: Additional compared approaches for the recommender system application.**
>
> **A2:** Yes, adding more compared approaches will greatly strengthen the paper. Therefore, we add the following compared approaches 1) Binary Cross Entropy Loss (BCE), which uses the assigned hard labels as the target and applies the cross entropy loss; 2) Bayesian Personalised Ranking (BPR) (Rendle et al., 2009), which uses the logistic loss to rank a pair of items; 3) Margin Ranking Loss (MR), which ranks a pair of items by using the margin loss. The hyperparameters are the same as those in Appendix I. The experimental results are summarised as follows:
>
> | Method        | HR    | NDCG  |
> | ------------- | ----- | ----- |
> | BCE           | 0.469 | 0.283 |
> | BPR           | 0.464 | 0.256 |
> | MR            | 0.476 | 0.271 |
> | Pcomp-Teacher | 0.179 | 0.066 |
> | Pconf         | 0.534 | 0.380 |
> | ConfDiff-ABS  | 0.570 | 0.372 |
>
> It is worth noting that BCE did not perform well because the heuristic labeling method can introduce a lot of label noise. Based on the experimental results, we can see that ConfDiff-ABS achieves the best performance in terms of HR and achieves comparable performance to Pconf. This confirms the effectiveness of our method in tackling this problem in recommender systems.
> ***
> **Q3: Experiments on the medical data sets.**
>
> **A3:** We agree that adding additional experiments on the medical application benchmarks will better motivate the setting. In the introduction, we pointed out a promising application of our setting and method in the medical domain. However, collecting data for medical applications is demanding because it requires a lot of domain knowledge and may involve privacy-sensitive issues. Therefore, we have not yet found such public data sets. We will consider collecting similar data sets as our future work. For example, we will consider collecting such data sets using a crowdsourcing platform (e.g. MTurk) by asking annotators to give the confidence difference between two examples. We are very grateful for your valuable suggestions.
>
> ***
> Reference:
>
> - Rendle et al., BPR: Bayesian personalized ranking from implicit feedback, UAI 2009.

---

> > ### Comment · Reviewer_Qs9H · 2023-08-16
> > **Follow up on Rebuttal**
> >
> > I would like to thank authors for their thorough responses and addressing my concerns.
> > I am changing my final score to 6.

---

### Author Rebuttal · Authors · 2023-08-09

First of all, we sincerely thank all the reviewers for their great efforts in reviewing this submission and providing helpful and valuable comments. Since we cannot revise our paper during the rebuttal period, we plan to make the following revisions in our paper:

- According to Reviewer Qs9H and Reviewer DX8U, we will include the performance of the supervised learning method in Tables 1 and 2.
- According to Reviewer Qs9H, we will include the results of additional compared approaches for the recommender system problem.
- According to Reviewer XYQ6, we will introduce a reference to the details of data sets in the paper.
- According to Reviewer G7aZ, we will add the details of the experimental setting for the Pcomp methods and the facts about Equations (6) and (8) to the paper. We will also discuss more about our data distribution assumption.

Besides, as suggested by Reviewer Qs9H and Reviewer XYQ6, we will consider collecting more real-world data sets for our problem as future work.

---

### Decision · Program_Chairs · 2023-09-21

**Decision:**

Accept (poster)

**Comment:**

Reviewers were unanimously supportive of this paper, which provides a consistent algorithm for learning from pairs of samples which are annotated with the difference of their probability of being positive. One recurring comment was that the practical applications of this framework are a little unclear; this appears to be fair comment that the authors may wish to remedy. Nonetheless, as a formal study of a statistical learning problem, the paper still appears to be above the acceptance threshold.